# Kindlin2 enables EphB/ephrinB bi-directional signaling to support vascular development

Wenqing Li[1],*, Lai Wen[2],* , Bhavisha Rathod[3], Anne-Claude Gingras[3,4] , Klaus Ley[2], Ho-Sup Lee[1]

**Direct contact between cells expressing either ephrin ligands or Eph receptor tyrosine kinase produces diverse developmental responses. Transmembrane ephrinB ligands play active roles in transducing bi-directional signals downstream of EphB/ephrinB interaction. However, it has not been well understood how ephrinB relays transcellular signals to neighboring cells and what intracellular effectors are involved. Here, we report that kindlin2 can mediate bi-directional ephrinB signaling through binding to a highly conserved NIYY motif in the ephrinB2 cytoplasmic tail. We show this interaction is important for EphB/ephrinB-mediated integrin activation in mammalian cells and for blood vessel morphogenesis during zebrafish development. A mixed two-cell population study revealed that kindlin2 (in ephrinB2-expressing cells) modulates transcellular EphB4 activation by promoting ephrinB2 clustering. This mechanism is also operative for EphB2/ephrinB1, suggesting that kindlin2-mediated regulation is conserved for EphB/ephrinB signaling pathways. Together, these findings show that kindlin2 enables EphB4/ephrinB2 bi-directional signal transmission.**

## Introduction

Transcellular interactions of erythropoietin-producing hepatoma (Eph) receptor tyrosine kinase with their ephrin ligands direct diverse cellular processes such as cell migration, axon guidance, and cell repulsion during morphogenesis and adult tissue homeostasis. These cell–cell contact-dependent mechanisms generate bi-directional signals: forward signals to Eph receptor–bearing cells and reverse signals to ephrin ligand–bearing cells. Eph receptors form two structurally distinct subfamilies, EphA and EphB, based on preferential association with glycophosphatidylinositol-linked ephrinA ligands and transmembrane ephrinB ligands, respectively. Nine EphA receptors that bind to five ephrinA ligands and five EphB

receptors that bind to three ephrinB ligands have been identified in the human genome (Lisabeth et al, 2013).

In response to ephrin engagement, the tyrosine kinase domain of Eph receptors transduces forward signaling by autophosphorylation and phosphorylation of effector proteins (Lisabeth et al, 2013; Kania & Klein, 2016) (Fig 1A). The 80-residue cytoplasmic tail of ephrinB ligand has high homology across species (Flanagan & Vanderhaeghen, 1998) and is essential for support of EphB signaling in vascular morphogenesis, angiogenesis, and lymphangiogenesis (Adams et al, 2001; Sawamiphak et al, 2010; Wang et al, 2010). Most ephrinB cytoplasmic domain–binding proteins, such as DVL2 and PDZ-RGS3, interact with its carboxy-terminal PDZ-binding motif (Lu et al, 2001; Mao et al, 2011). However, the mechanism by which ephrinB cytoplasmic domain–binding proteins enable forward signaling to EphB-expressing cells remains incompletely understood and the molecular players that direct this process are still unknown.

The kindlins play essential roles in integrin activation by binding to a highly conserved membrane-distal NxxY motif in the integrin $\beta$ cytoplasmic tail (Harburger et al, 2009). The kindlin family consists of three members (kindlin1, kindlin2, and kindlin3), each having a 4.1 protein, ezrin, radixin, and moesin (FERM) domain, and a PH domain. Kindlin2 is broadly expressed, except in blood cells, whereas kindlin1 and kindlin3 are primarily expressed in epithelial cells and hematopoietic cells, respectively (Rognoni et al, 2016). Inactivation of kindlin2 gene in mice leads to peri-implantation lethality (Dowling et al, 2008; Montanez et al, 2008). Kindlin2 is essential for angiogenesis during vascular development in zebrafish (Pluskota et al, 2011).

Here, we identify kindlin2 as an ephrinB2-interacting protein. We show that kindlin2 interacts with the ephrinB2 cytoplasmic tail through an NIYY motif that is conserved among B ephrins and that this interaction is involved in arteriovenous segregation in zebrafish. In mixed two-cell population experiments, kindlin2 binding to ephrinB2 is required for EphB4 activation in opposing cells. This kindlin2-mediated regulation is conserved for EphB2/ephrinB1, another EphB/ephrinB pair. Finally, we establish that kindlin2 binding to ephrinB2 supports ephrinB2 clustering and cell spreading. We propose that this clustering mediates both forward

[1]Department of Medicine, University of California San Diego, La Jolla, CA, USA   [2]La Jolla Institute for Immunology, La Jolla, CA, USA   [3]Lunenfeld-Tanenbaum Research Institute, Mount Sinai Hospital, Sinai Health System, Toronto, Canada   [4]Department of Molecular Genetics, University of Toronto, Toronto, Canada

Correspondence: hol001@health.ucsd.edu
*Wenqing Li and Lai Wen contributed equally to this work

**A**

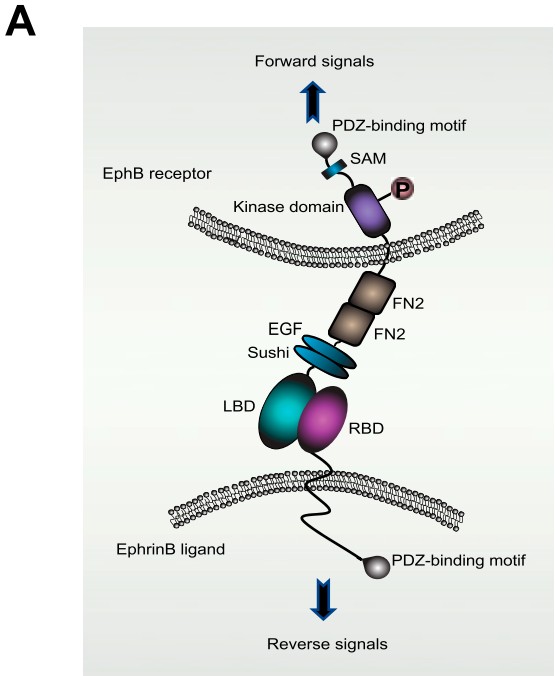

**B** **C**

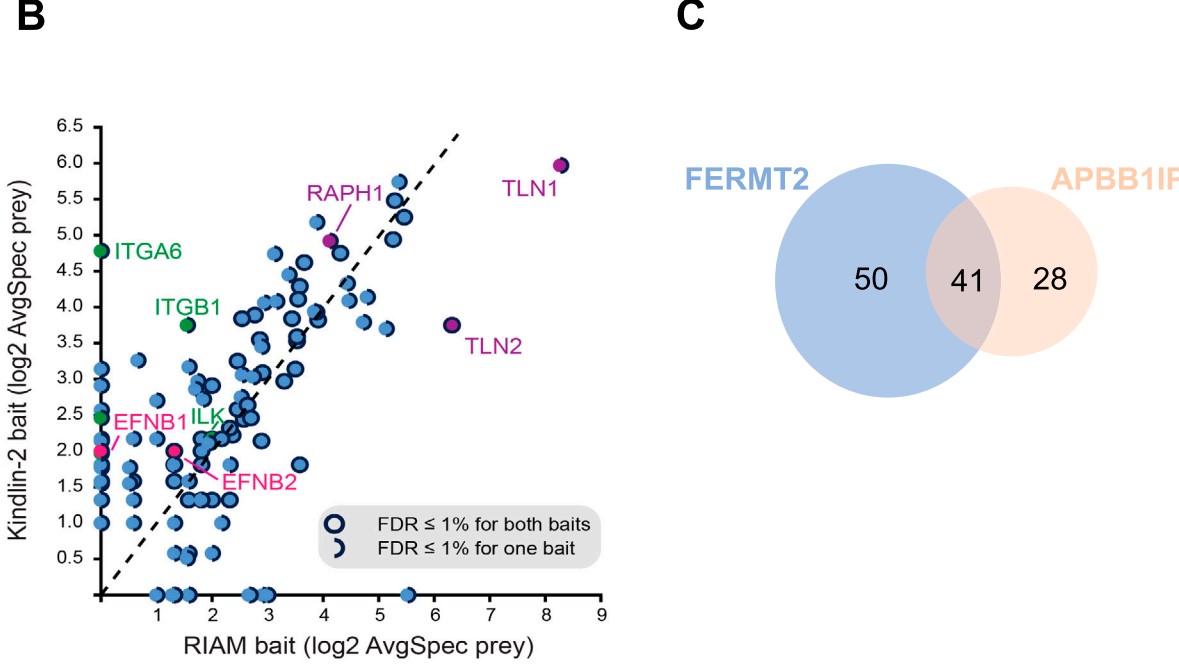

**Figure 1. BioID of kindlin2 and RIAM baits.**
**(A)** Domain structure and bi-directional signaling of ephrinB and EphB. EphB receptors and ephrinBs located at the cell membrane of opposing cells interact in trans and activate short-distance bi-directional signaling. LBD, ligand-binding domain; RBD, receptor-binding domain; FN, fibronectin domain; EGF, epidermal growth factor–like domain; SAM, sterile alpha motif. Orange circle indicates tyrosine phosphorylation. **(B)** Scatterplot comparing the averaged spectra (log$_2$-transformed) for high-confidence preys identified from kindlin2 with RIAM baits; notable preys are indicated and filled in different colors, with details in the legend inset. **(C)** Venn diagram showing the number of overlapped preys discovered in the proximity interactomes of kindlin2 and RIAM baits. All high-confidence interactions (AvgP ≥ 0.95) for each bait were considered. See Fig S1 for details.

signaling to EphB4-expressing cells and reverse signaling into ephrinB2-expressing cells. In sum, these data establish that kindlin2 binding to B ephrins drives the clustering of those receptors and plays an essential bi-directional EphB/ephrinB signaling that enables vascular development and potentially other EphB/ephrinB-regulated developmental processes.

# Results

## Proximity-dependent biotinylation identifies ephrinB as a kindlin2-proximal protein

To find novel regulators involved in kindlin2-dependent integrin signaling, we set out to perform unbiased proximity-dependent biotinylation labeling (BioID) and identification by mass spectrometry. We generated tetracycline-inducible Flp-In T-REx HEK293 cell pools expressing BirA*-FLAG-kindlin2 (gene name FERMT2) or BirA*-FLAG-RIAM (gene name APBB1IP, another regulator of integrin activation); addition of biotin alongside tetracycline for 24 h was carried out to mediate biotinylation of proximal partners of kindlin2 or RIAM. To uncover proteins specifically enriched with kindlin2 or RIAM, negative controls consisting of cells expressing BirA*-FLAG alone or fused to the GFP were used for scoring using SAINTexpress (Teo et al, 2014) software, and high-confidence proximal interactions were defined as those ≤ 1% FDR (Fig 1B; some proteins are specifically enriched with kindlin2 over RIAM and vice versa). Kindlin2 and RIAM BioID revealed 91 and 69 high-confidence proximity interactors, respectively, with 41 overlapping interactions (Figs 1C and S1). Pathway enrichment analysis using g:Profiler shows that most of these kindlin2-proximal proteins are enriched in anchoring junction, cell adhesion molecule binding, membrane, and actin filament–based processes, respectively (Fig S2), validating identification of biologically relevant proteins by kindlin2 bait. Kindlin2 BioID includes known direct interactors such as integrins and ILK (Fig 1B, green labels). RIAM-proximal interactors also contain known interactors, including talin1 and talin2 (Fig 1B, purple). However, there are ~50 significant and previously unrecognized interactors for kindlin2 that are preferentially enriched to kindlin2 interactome. We notably identified ephrinB family of proteins as kindlin2-proximal interactors (Fig 1B, pink), suggesting a potential new physical connection.

## Kindlin2 interacts with ephrinB2 through an NIYY motif in the ephrinB2 cytoplasmic tail

To validate the potential interaction between kindlin2 and ephrinB2 and localize the specific kindlin2-binding site, we examined the carboxy-terminal amino acid sequences of ephrinB2. We noticed that the ephrinB2 cytoplasmic tail contains an NIYY motif next to its PDZ-binding motif (Fig 2A, upper panel). This motif, together with an upstream serine residue, is similar to the consensus membrane-distal kindlin2-binding NxxY motif and adjacent serine or threonine residues in integrin cytoplasmic tails. This ephrinB2 NIYY motif is conserved from human to zebrafish (Fig 2A, lower panel).

To determine whether kindlin2 binds ephrinB2 through the NIYY motif, we performed affinity chromatography using the cytoplasmic tails of ephrinB2 WT, ephrinB2 mutant in which the NIYY motif was replaced by four alanine residues (4A), integrin β1A, or integrin αIIb, each immobilized on NeutrAvidin beads. After incubation with a cell lysate containing GFP-kindlin2, the ephrinB2 WT tail, but to a much lesser extent the 4A mutant, was able to interact with kindlin2 (Fig 2B). The fraction bound was comparable to that obtained with integrin β1A. Thus, kindlin2 interacts with ephrinB2 primarily through the NIYY motif of the ephrinB2 cytoplasmic tail.

We next focused on the consensus YKV PDZ-binding motif at the carboxy-terminus of ephrinB2. This PDZ-binding motif is highly conserved in the ephrinB family and mediates ephrinB reverse signaling (Kania & Klein, 2016). The kindlin2-binding NIYY motif partially overlaps with the consensus YKV PDZ-binding motif; therefore, we tested whether the 4A mutant that blocked kindlin2 binding to the NIYY motif disrupted the interaction with the PDZ protein DVL2. Affinity chromatography revealed that the 4A mutant ephrinB2 tail bound DVL2 to a similar extent as the WT ephrinB2 tail (Fig 2C). Thus, the ephrinB2 4A mutant disrupts kindlin2 binding but not the binding of a PDZ domain protein, DVL2.

To assess whether the ephrinB2 tail directly binds kindlin2, we used in vitro protein interaction assay. We purified kindlin2 from mammalian cells and performed affinity chromatography with the purified cytoplasmic tails of ephrinB2 WT, ephrinB2 mutant (4A), or integrin αIIb as a negative control (Fig 2D). This assay reveals that the cytoplasmic tail of ephrinB2 directly associated with kindlin2, whereas the tail of ephrinB2 kindlin2-binding mutant (4A) bound with much reduced affinity. Direct association between kindlin2 and ephrinB2 through an NxxY motif in the cytoplasmic tail prompted us to identify amino acid residues important for the binding. We therefore mutated the residues in the NxxY motif and upstream serine residue that is critical for kindlin2 binding in the integrin cytoplasmic tail (Harburger et al, 2009) and tested their effect on kindlin2 binding in vitro (Fig 2E and F). Point mutations in S325A or N328A did not substantially inhibit kindlin2 binding. Y331A and a double point mutation (2A, N328A and Y331A) partially inhibited the kindlin2 binding. In contrast, 4A mutation (N328A, I329A, Y330A, and Y331A) strongly impaired the direct binding to kindlin2. Together, these results indicate that IYY residues in the NIYY motif of ephrinB2 are required to support direct interaction with kindlin2.

Kindlin has a distinct FERM (4.1 protein, ezrin, radixin, and moesin) domain composed of three lobes (F1, F2, and F3) where the PH domain is inserted within the F2 lobe. Kindlin interacts with integrin β tails through the F3 lobe. We tested whether binding to ephrinB2 is mediated by the F3 subdomain using integrin-binding defective mutant (W615A) of kindlin2 (Harburger et al, 2009). We found the interaction with ephrinB2 was not significantly affected by the mutation (Fig 2G and H), suggesting that other kindlin2 domains are likely involved. This result indicates that kindlin2 binding to ephrinB2 and to integrin is not mutually exclusive.

The ephrinB family has two other members, ephrinB1 and ephrinB3, and they share high sequence homology in their carboxy-termini (Fig 2J). We therefore assessed whether ephrinB1 or ephrinB3 can interact with kindlin2. As expected, pulldown assay using cytoplasmic tails of ephrinB1 and ephrinB3 demonstrated interaction with kindlin2, whereas mutation of the NIYY motifs to alanine (4A) abolished the interaction (Fig 2I).

Because Eph/ephrin signaling plays critical roles in hematopoiesis (Tosato, 2017), we asked whether kindlin3 that is primarily expressed in hematopoietic cells has the ability to bind ephrinB2. Affinity chromatography using the cytoplasmic tails of ephrinB2 WT, ephrinB2 kindlin2-binding mutant (4A), integrin β1A, or integrin αIIb, each immobilized on NeutrAvidin beads, with a cell lysate containing kindlin3 showed that the ephrinB2 WT tail was able to interact with kindlin3, whereas the 4A mutant binds with a reduced affinity (Fig 2K). Thus, kindlin3 is able to interact with ephrinB2 through the NIYY motif in the ephrinB2 cytoplasmic tail.

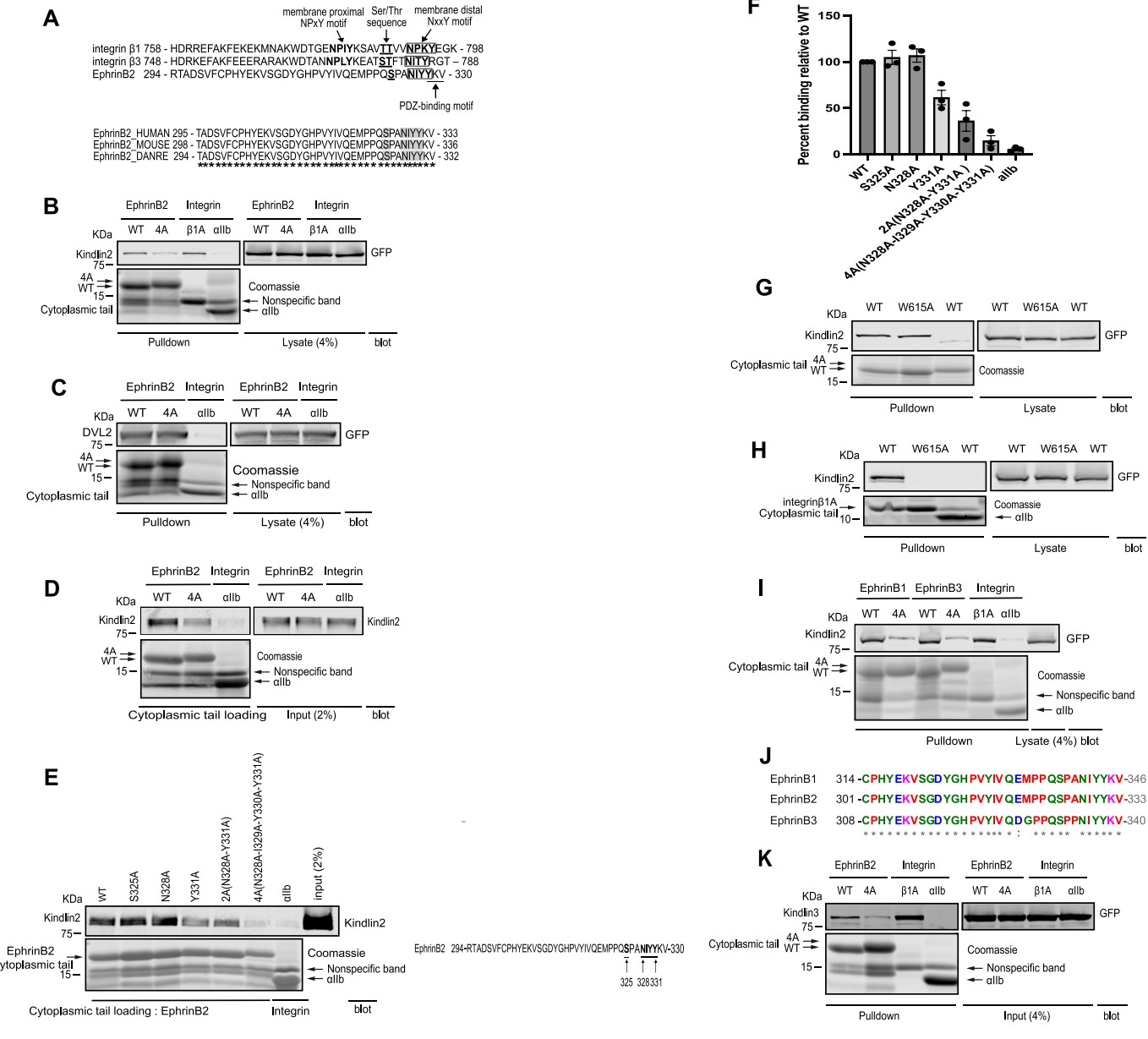

**Figure 2. Kindlin2 binds NxxY motif in the ephrinB2 cytoplasmic tail.**
**(A)** Cytoplasmic tail sequence alignment compares ephrinB2 with integrins β1 and β3 (upper panel). Membrane-proximal NPxY motif in integrins β1 and β3 is bolded. Serine/threonine residues are bolded and underlined. NxxY motif is bolded and boxed, and ephrinB2 PDZ-binding motif is underlined. Conservation of ephrinB2 Ser/Thr residue and NxxY motif in human, mouse, and zebrafish was examined using Clustal Omega program (http://www.ebi.ac.uk/Tools/msa/clustalo/) (lower panel). "*" indicates fully conserved amino acid residue. **(B)** Pulldown assay using purified his/Avi-tagged cytoplasmic tails of ephrinB2 WT, kindlin2-binding mutant 4A (NxxY to four alanine mutants), and integrin β1A or αIIb immobilized on NeutrAvidin beads was performed with a cell lysate expressing GFP-kindlin2. Bound proteins were washed and separated on SDS–PAGE, and binding of kindlin2 was detected by Western blot with anti-GFP antibody. Immobilized cytoplasmic tails were visualized by Coomassie brilliant blue staining. Both WT and 4A mutant were expressed equally well. **(C)** Pulldown assay of his/Avi-tagged cytoplasmic tails of ephrinB2 WT, 4A, or integrin αIIb immobilized on NeutrAvidin beads with a cell lysate expressing GFP-Dvl2. Associated proteins were subjected to SDS–PAGE and subsequent Western blotting. Dvl2 was recognized by an anti-GFP antibody. Immobilized cytoplasmic tails were stained by Coomassie brilliant blue. **(D)** In vitro binding assay mapping the kindlin2-binding site on the ephrinB2 cytoplasmic tail. Purified kindlin2 was mixed with cytoplasmic tails of ephrinB2 WT, kindlin2-binding mutant (4A), or integrin αIIb immobilized on NeutrAvidin beads and assessed for binding. Anti-kindlin2 antibody was used to detect bound kindlin2 in Western blot, and cytoplasmic tails were stained by Coomassie brilliant blue. **(E)** In vitro binding assay between purified kindlin2 and cytoplasmic tails of ephrinB2 with mutations of critical residues in NxxY motif and upstream serine residue immobilized on NeutrAvidin beads (left panel). The mutated residues were indicated in the ephrinB2 cytoplasmic tail sequence (right panel). Bound kindlin2 was recognized by anti-kindlin2 antibody in Western blot, and cytoplasmic tails were detected by Coomassie brilliant blue. **(F)** Binding of kindlin2 was quantified by densitometry, normalized to cytoplasmic tail loading, and then calculated relative to the binding of ephrinB2 WT (means ± SEM; n = 3). **(G)** Pulldown analysis of his/Avi-tagged cytoplasmic tails of ephrinB2 WT or 4A as a negative control, immobilized on NeutrAvidin beads with a cell lysate expressing GFP-kindlin2 WT or W615A, respectively. Bound proteins were fractionated on SDS–PAGE and probed with an anti-GFP antibody in Western blot. Immobilized cytoplasmic tails were visualized by Coomassie brilliant blue staining. **(H)** Pulldown of his/Avi-tagged cytoplasmic tails of integrin β1A or αIIb on NeutrAvidin beads with a cell lysate expressing GFP-kindlin2 WT or W615A, respectively. Pulldown proteins were washed and separated on SDS–PAGE. GFP-kindlin2 was recognized by anti-GFP antibody in Western blot. Purified

## Kindlin2 co-localizes with ephrinB2 in mammalian cells

To determine the localization of ephrinB2 during mammalian cell adhesion to EphB4, we co-infected NIH3T3 cells with lentiviruses encoding mCherry-kindlin2 in combination with EGFP-ephrinB2 or EGFP-ephrinB2 kindlin2-binding mutant (4A). We used total internal reflection fluorescence (TIRF) microscopy to localize ephrinB2 and kindlin2 at the basal plasma membrane of the cells with submicron resolution. EGFP-ephrinB2 co-localized with kindlin2 in clusters at the ventral surface (Fig 3A and B). In contrast, the kindlin2-binding mutant (4A) was diffusely localized at the cell periphery and showed no co-localization with kindlin2. The expression of EGFP-ephrinB2 or mutant (4A) did not alter the formation or distribution of focal adhesions as evidenced by analyzing paxillin-containing focal adhesions under an TIRF microscope (Fig S3). Furthermore, we validated the interaction between ephrinB2 and kindlin2 in mammalian cells by co-immunoprecipitation. When mCherry-kindlin2 was co-expressed with flag-ephrinB2 or flag-kindlin2–binding mutant (4A) in HEK293 cells, immunoprecipitation of flag-ephrinB2 with anti-flag antibody captured bound kindlin2, whereas immunoprecipitation of kindlin2-binding mutant (4A) did not (Fig 3C). Thus, the NIYY motif mediates the interaction of full-length ephrinB2 with kindlin2 in mammalian cells, thereby accounting for the proximity of the two proteins revealed by BioID.

## The binding of kindlin2 to ephrinB2 mediates EphB4/ephrinB2-dependent integrin activation

Previous works suggested that Eph receptors and ephrin ligands play important roles in integrin-dependent cell adhesion, which leads to increased or decreased adhesion, depending on cell context (Huynh-Do et al, 1999; Zou et al, 1999; Hamada et al, 2003; Meyer et al, 2005; Miao et al, 2005; Foo et al, 2006; Singh et al, 2012; Zhang et al, 2015). To determine whether EphB4 and ephrinB2 support integrin activation, we used HEK293 cells expressing a recombinant talin-dependent activated integrin $\alpha IIb(R995A)\beta 3$ and assessed integrin activation by the binding of a ligand-mimetic antibody, PAC1 (O'Toole et al, 1994). The HEK293 cells express endogenous ephrinB2 and EphB4 and grow in clusters with cell–cell contacts (Fig S4). Silencing the expression of endogenous ephrinB2 led to decreased integrin activation (Fig 4A and B). Integrin activation was rescued by the expression of shRNA-resistant ephrinB2 but not by the kindlin2-binding defective mutant ephrinB2 (4A) (Fig 4A and B).

We tested whether EphB4, the binding receptor of the ephrinB2 ligand, plays a role in integrin activation as well. When we rescued the ephrinB2-silenced cells with a mutant of the extracellular domain in ephrinB2 (6A, F117A, P119A, L121A, W122A, L124A, and F126A) that is unable to bind to EphB4, we failed to rescue integrin

activation (Fig 4A and B). The failure of this mutant to bind EphB4 was confirmed by co-immunoprecipitation (Fig 4C). Similarly, silencing of endogenous EphB4 expression resulted in reduced integrin activation (Fig 4D and E) and the expression of shRNA-resistant EphB4 rescued the integrin activation defect. Importantly, a rescue construct expressing an EphB4 kinase-dead mutant did not rescue the integrin activation defect. In sum, these data show that the association of ephrinB2 with kindlin2 is required for EphB4 kinase–dependent integrin activation.

Furthermore, to directly address whether ephrinB2 and EphB4 play a role in other integrin signaling, we silenced the expression of ephrinB2 or EphB4 in WT HEK293 cells and assessed binding of 9EG7 antibody, which specifically reports integrin $\beta 1$ activation (Bazzoni et al, 1995). Silencing ephrinB2 or EphB4 expression significantly decreased 9EG7 binding to integrin $\beta 1$ (Fig S5) that indicates ephrinB2 and EphB4 support integrin $\beta 1$ activation.

## Kindlin2/ephrinB2 interaction is essential for vascular development

EphrinB2 regulates both vasculogenesis and angiogenesis during development (Wang et al, 1998; Adams et al, 1999; Gerety et al, 1999), and kindlin2 is involved in angiogenesis (Pluskota et al, 2011). To uncover the potential role of kindlin2/ephrinB2 interaction during vascular development, we used zebrafish as a model system. We first confirmed zebrafish protein interaction between ephrinB2a and kindlin2 by affinity chromatography using immobilized cytoplasmic tails of ephrinB2a WT, ephrinB2a kindlin2-binding mutant (4A), integrin $\beta 1A$, or integrin $\alpha IIb$ with a cell lysate from cells expressing kindlin2. The cytoplasmic tail of ephrinB2a bound kindlin2, whereas ephrinB2a (4A) lacking the kindlin2-binding NIYY motif did not (Fig 5A). This result demonstrates zebrafish kindlin2 binds ephrinB2a, and this binding is dependent on the NIYY motif in the ephrinB2a cytoplasmic tail.

Next, we first employed morpholino-induced depletion of *efnb2a* and subsequent rescue by *efnb2a* WT or kindlin2-binding mutant (4A) mRNA to test the role of the ephrinB2a/kindlin2 interaction in arteriovenous segregation. Remarkably, administration of *efnb2a* mRNA resulted in defective circulation of red blood cells and obstruction of the dorsal aorta (Video 1) and a shortened loop (Video 2) similar to defects observed after administration of *efnb2a* morpholino. *Efnb2a* expression is normally limited to the arterial endothelial cells while absent in the venous endothelial cells, and mediates arteriovenous segregation by interaction with ephb4 expressed in venous endothelial cells (Wang et al, 1998). We reasoned that an ectopic expression of *efnb2a* in venous angioblasts, by scrambling efnb2-ephb4 signaling, impaired the segregation of arterial and venous endothelial cells. This hypothesis is supported by a previous observation that application of recombinant

---

cytoplasmic tails on NeutrAvidin beads were visualized by Coomassie brilliant blue staining. **(I)** His/Avi-tagged cytoplasmic tails of ephrinB1, ephrinB3, and integrin $\beta 1$ or $\alpha IIb$ immobilized on NeutrAvidin beads were mixed with a cell lysate expressing GFP-kindlin2. The pulldown results were visualized by SDS–PAGE and Western blotting with anti-GFP antibody. All mutants were expressed equally well as WT. **(J)** Protein alignment (right panel) of cytoplasmic domains ephrinB1, ephrinB2, and ephrinB3 was generated by Clustal Omega. "*" denotes fully conserved amino acid residue, and ":" indicates conserved residue with similar properties among ephrinB cytoplasmic tails. **(K)** Pulldown of kindlin3 with his/Avi-tagged cytoplasmic tails of ephrinB2 WT, 4A, and integrin $\beta 1A$ or $\alpha IIb$ immobilized on NeutrAvidin beads. Bound proteins were separated on SDS–PAGE. Cytoplasmic tails were stained by Coomassie brilliant blue staining, and kindlin3 was probed by anti-GFP antibody in Western blot. Source data are available for this figure.

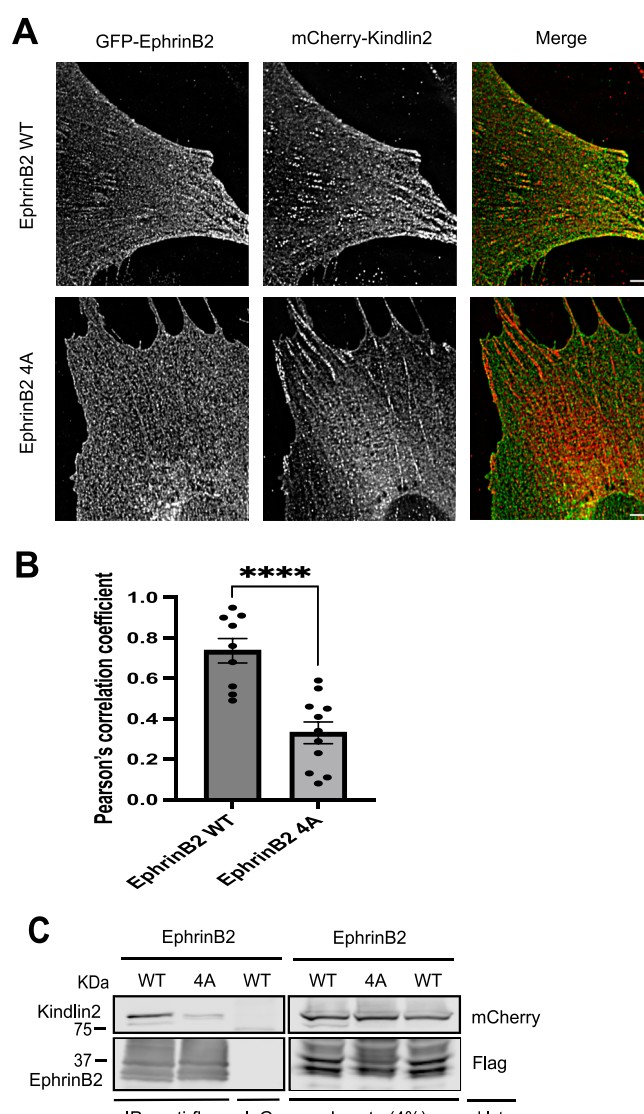

**Figure 3. Kindlin2 and ephrinB2 are associated in mammalian cells.**
**(A)** Kindlin2 co-localizes with ephrinB2 in adhesive structures. NIH3T3 cells were co-transduced by lentivirus particles expressing mCherry-kindlin2 with GFP-ephrinB2 WT or 4A, respectively. The cells were plated on laminin-coated wells. Localization of the two proteins was visualized by an TIRF microscope. Scale bar, 6 μm. **(B)** Co-localization of kindlin2 with ephrinB2 WT or 4A was measured as the mean values of Pearson's correlation coefficient on TIRF images. EphrinB2 WT, n = 9; EphrinB2 4A, n = 11. The data are presented as the mean ± SEM. Statistical analysis used an unpaired two-tailed *t* test. ****P < 0.0001. **(C)** mCherry-kindlin2 was co-transfected into HEK293 cells with flag-ephrinB2, flag-ephrinB2 (4A), or flag-integrin *β*1 as a positive control. The cells were lysed, and flag-tagged proteins were immunoprecipitated by anti-flag antibody or control IgG as a negative control. The immunocomplexes were washed and separated on SDS–PAGE. Associated kindlin2 was recognized by anti-mCherry antibody in Western blot. All proteins were expressed equally well. Source data are available for this figure.

EphrinB2a protein into chick embryo led to the formation of arterial/venous shunts (le Noble et al, 2004). Indeed, further examination of the vessels showed that *efnb2a* overexpression led to a narrow aorta, shorter intersegmental vessel (Fig 5C and Video 1), or arterial/venous shunt (Fig 5F and Video 2) compared with those

of uninjected embryos (Fig 5E and H and Video 3). In contrast to the effect of mis-expressing WT *efnb2a*, these defects were absent in kindlin2-binding mutant *efnb2a* 4A (N327A, I328A, Y329A, and Y330A)–injected embryos (Fig 5D and G and Video 4 and Video 5). Total expression of *efnb2a* WT and kindlin2-binding mutant (4A) was confirmed in Western blots of the injected embryos (Fig S6A and B). The cell surface protein expression of *efnb2a* WT and mutant (4A) was verified by flow cytometry (Fig S6C). Thus, the kindlin2-binding NIYY motif of efnb2a is essential for its capacity to support arteriovenous segregation during vascular development.

## Kindlin2 association with ephrinB2 regulates EphB4 forward signaling

The foregoing studies established that kindlin2 binding to ephrinB2 is required for integrin function and vascular development. To further investigate the roles of kindlin2 in the EphB4/ephrinB2 pathway and to mimic the EphB4/ephrinB2 system, we established a co-culture system using two stable cell lines (Fig 6A), BT16 expressing flag-ephrinB2 or flag-ephrinB2 (4A), and HEK293 expressing HA-EphB4. The BT16 cell line was retrieved from the DepMap online tool (https://depmap.org) because BT16 cells do not express ephrinB2 (Fig S4) and kindlin2 is a primary member of the kindlin family expressed in the cells (https://depmap.org).

We co-cultured the two cell lines and used an anti-HA antibody affinity column to purify EphB4/ephrinB2 complex and measured EphB4 phosphorylation to assess EphB4 activation. We first examined whether kindlin2 association with ephrinB2 is required for EphB4 activation. EphB4 in HEK293 cells was autophosphorylated when incubated with BT16 cells expressing ephrinB2 WT, whereas the phosphorylation was diminished when co-cultured with BT16 cells expressing ephrinB2 (4A) (Fig 6B–D). Furthermore, we down-regulated endogenous kindlin2 expression in BT16 cells by RNAi-mediated gene silencing and measured EphB4 autophosphorylation in HEK293 cells. EphB4 phosphorylation was reduced when kindlin2 expression was silenced in BT16 cells (Fig 6E–H). These results indicate kindlin2 interaction with ephrinB2 enables EphB4 forward signaling.

In the previous study, we showed EphB4 activation is necessary for EphB4-dependent integrin activation. We then asked whether integrin function in EphB4-expressing cells could be related to EphB4 forward signaling in the co-culture system. We silenced integrin *β*1 expression in HEK293 cells and incubated with BT16 cells expressing ephrinB2 to assess EphB4 phosphorylation. EphB4 phosphorylation was reduced compared with that in cells treated with control shRNA (Fig 6I–L). These data demonstrate integrin function is required for EphB4 forward signaling.

EphrinB family shares very high amino acid sequence similarity in their cytoplasmic tails, and our studies show kindlin2 binds cytoplasmic tails of all three EphrinB family members (Fig 2G and H). To investigate whether regulation of EphB forward signaling by kindlin2/ephrinB association is a common mechanism in EphB/ephrinB family, we evaluated ligand ephrinB1 receptor EphB2 signaling. We generated stable cell lines, BT16 expressing flag-ephrinB1 or flag-ephrinB1 kindlin2-binding mutant (4A), and HEK293 expressing HA-EphB2, and performed co-culture analyses

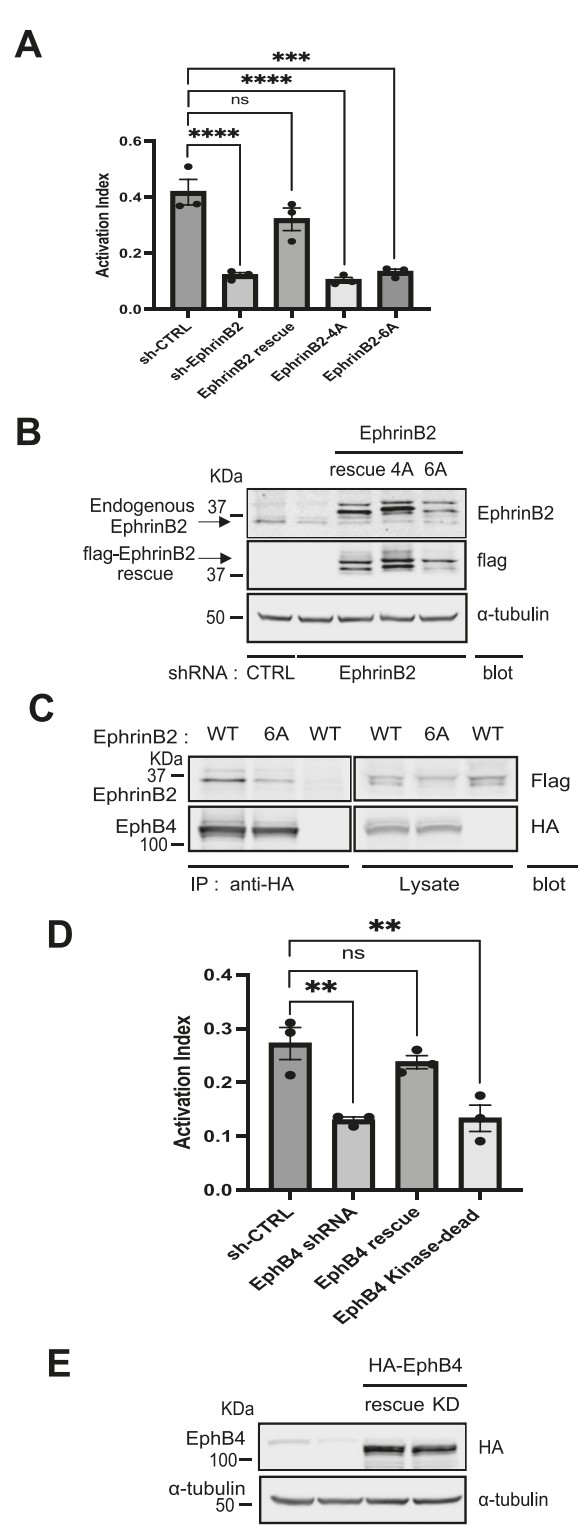

**Figure 4.  EphrinB2 and EphB4 regulate integrin signaling.**
**(A)** Silencing ephrinB2 expression resulted in integrin activation defect. The expression of endogenous ephrinB2 was silenced by lentivirus-based RNAi in 293 cells expressing constitutively active αIIb(R995A)β3. A scrambled shRNA was used as a negative control. **(C)** For rescue experiment, cells were transiently transfected with cDNAs encoding empty vector, shRNA-resistant ephrinB2,

to assess EphB2 activation. We found co-culturing the two cells induced EphB2 phosphorylation, whereas ephrinB1 kindlin2-binding mutant was not able to induce EphB2 phosphorylation in HEK293 cells (Fig 6M–O). Thus, the regulation of EphB forward signaling by kindlin2 is conserved in EphrinB family.

## Kindlin2 mediates ephrinB2 reverse signaling

To activate Eph receptors, ephrin ligands must be in a clustered or membrane-associated form to facilitate dimerization or aggregation. In contrast, soluble monomeric ligands do not induce the EphB receptor autophosphorylation (Davis et al, 1994). For EphA receptors, soluble monomeric ephrinA1 ligand has been reported to activate EphA2 and other EphA receptors (Wykosky et al, 2008). Kindlin has been shown to increase the ability of integrin to bind multivalent ligands by promoting talin-activated integrin clustering (Ye et al, 2013). Therefore, we investigated whether kindlin2 promotes clustering of ephrinB2. We used TIRF microscopy that can visualize clusters at the submicron level. BT16 cells expressing ephrinB2 or ephrinB2 kindlin2-binding mutant (4A) were plated on a well coated with EphB4-Fc to activate ephrinB2 signaling through the cytoplasmic domain (Palmer et al, 2002) and laminin to promote the adhesion of BT16 cells because EphB4-Fc alone does not support cell adhesion. BT16 cells expressing ephrinB2 kindlin2-binding mutant have significantly fewer number of clusters than cells expressing ephrinB2 (Fig 7A and B), and brightness and size of EphrinB2 punta were also reduced, even though equal amounts of proteins were expressed in each cell (Fig 7E and F). This finding indicates kindlin2 promotes the clustering of ephrinB2 at the submicron scale.

Furthermore, we observed a trend toward a reduced size of kindlin2-binding mutant–expressing cells plated on an EphB4-Fc only–coated well, implying cell adhesion may be impaired. Hence, we assessed the ability of cells to adhere to a plate coated with

ephrinB2 (4A), or ephrinB2 (6A, EphB4-binding mutant, shown in (C)), respectively. The binding of PAC1 (activation-specific anti-αIIbβ3 monoclonal antibody) was assessed by FACS analysis. The activation index of transfected cells was calculated as $(F - Fo)/(Fm - Fo)$, in which $F$ is the median fluorescence intensity (MFI) of PAC1 binding; $Fo$ is the MFI of PAC1 binding in the presence of competitive inhibitor (Integrilin, 1 μM); and $Fm$ is the MFI of PAC1 binding in the presence of the integrin-activating antibody anti-LIBS6 (2 μM). Results represent the mean ± SEM (n = 3) (**$P$ < 0.01; one-way ANOVA) (left panel). **(B)** Expression of ephrinB2 and its mutants was confirmed by immunoblotting of SDS–PAGE-fractionated cell lysates. **(C)** Co-IP of EphB4 and ephrinB2. Cell lysates were produced from cells co-transfected with HA-EphB4 and flag-ephrinB2 WT or 6A and immunoprecipitated by anti-HA antibody. Captured immunocomplex was washed and separated on SDS–PAGE. Bound ephrinB2 was recognized in Western blot with anti-flag antibody. HA-EphB4 and flag-EphrinB2 WT or 6A were expressed equally well. **(D)** Knockdown of EphB4 expression caused a defect in integrin activation. Endogenous EphB4 expression was reduced by lentivirus shRNA in the 293(αIIb[R995A]β3) cells. And the cells were transiently transfected with cDNAs encoding empty vector, shRNA-resistant EphB4, or EphB4 kinase-dead mutant, respectively. Integrin activation was assessed by PAC1 (activation-specific anti-αIIbβ3 monoclonal antibody) on FACS analysis. **(A)** Activation index of transfected cells was calculated as in (A). Results represent the mean ± SEM (n = 3) (**$P$ < 0.01; one-way ANOVA). **(E)** Expression of EphB4 and its mutants was verified by Western blotting of SDS–PAGE-fractionated cell lysates.
Source data are available for this figure.

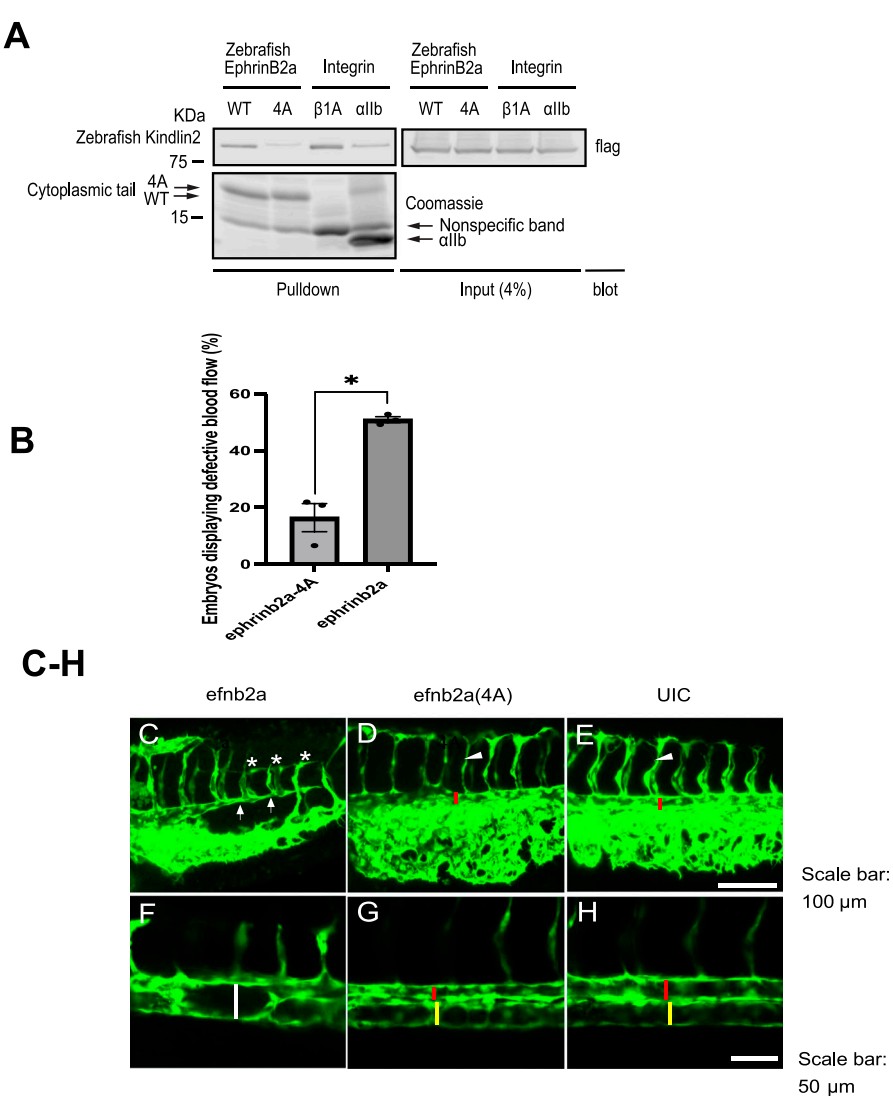

**Figure 5.** **Perturbation of kindlin2 binding to ephrinB2 causes defects in vascular development in zebrafish.**

**(A)** Pulldown experiment using cytoplasmic tails of zebrafish ephrinB2, ephrinB2 (4A), and integrin β1A or αIIb immobilized on NeutrAvidin beads with a cell lysate expressing GFP-zebrafish kindlin2. Bound proteins were fractionated on SDS–PAGE. His/Avi-tagged cytoplasmic tails were stained by Coomassie blue, and kindlin2 was recognized by Western blotting with anti-GFP antibody. Asterisk indicates the position of purified his/Avi-tagged cytoplasmic tails. **(B)** Percentage of zebrafish embryos injected with *efnb2a* WT or 4A RNAs displaying defects in blood flow. The data are presented as the mean ± SEM (n = 3). An unpaired two-tailed *t* test was used. **P < 0.01. **(C, D, E)** Some zebrafish embryos that overexpressed ephrinb2a displayed extremely narrow dorsal aorta (indicated by arrow) and inadequate intersegmental vessel growth (indicated by star), whereas embryos that overexpressed 4A mutant of ephrinb2a (D) and uninjected embryos (UIC) (E) did not show these defects (red bar indicated the 10 aorta, and arrowhead indicated the intersegmental vessel). **(F, G, H)** Other zebrafish embryos that overexpressed ephrinb2a failed the arterial/venous segregation (indicated by white bar) in local area of the trunk, whereas this defect was not seen in 4A-overexpressed embryos (G) and uninjected embryos (UIC) (H) (red and yellow bars indicated the dorsal aorta and posterior cardinal vein, respectively). Scale bar, 100 μm (C, D, E) and 50 μm (F, G, H).

Source data are available for this figure.

EphB4-Fc. We found BT16 cells expressing ephrinB2 kindlin2-binding mutant exhibited significantly reduced adhesion compared with cells expressing ephrinB2 (Fig 7C and D). Total expression and cell surface expression of ephrinB2 and mutant were verified by Western blot and flow cytometry, respectively (Fig 7E and F). Taken together, these results indicate kindlin2 facilitates ephrinB2 signaling by promoting ephrinB2 clustering and adhesion to EphB4.

## Discussion

In this study, we show that kindlin2 mediates bi-directional EphB4/ephrinB2 signaling through binding to a consensus motif in the ephrinB2 cytoplasmic tail. We demonstrate this binding is essential for integrin signaling in mammalian cells and vascular development in zebrafish. In mixed two-cell populations, we find that kindlin2 in ephrinB2-expressing cells modulates EphB4 activation in neighboring EphB4-expressing cells by promoting ephrinB2 clustering. Moreover, destroying kindlin2-binding site in ephrinB2 leads to a decrease in cell adhesion, suggesting kindlin2 is involved in reverse signaling.

Our proximity-dependent biotinylation labeling and comparative enrichment analysis with another integrin activation regulator, RIAM, revealed that ephrinBs and integrins are more enriched in kindlin2 adhesome than in RIAM adhesome, whereas talins are more enriched in RIAM adhesome. Our data supported that ephrinB family interacts with kindlin2 through a consensus NIYY motif in its cytoplasmic tail. This motif is highly conserved in ephrinB family, from zebrafish to mammals. Furthermore, the fact that ephrinB2 is also able to bind kindlin3 (Fig 2I) indicates ephrinB family is a novel interactor of the kindlin family.

Tyrosine phosphorylation of the ephrinB cytoplasmic tail has been shown to play important roles in ephrinB reverse signaling. Mutational analysis of the NIYY motif in the ephrinB2 tail shows mutation of the tyrosine residue in the NIYY motif (Y331A) partially affected binding to kindlin2, whereas 4A mutation significantly impaired the ability of ephrinB2 to bind kindlin2. Further analysis is

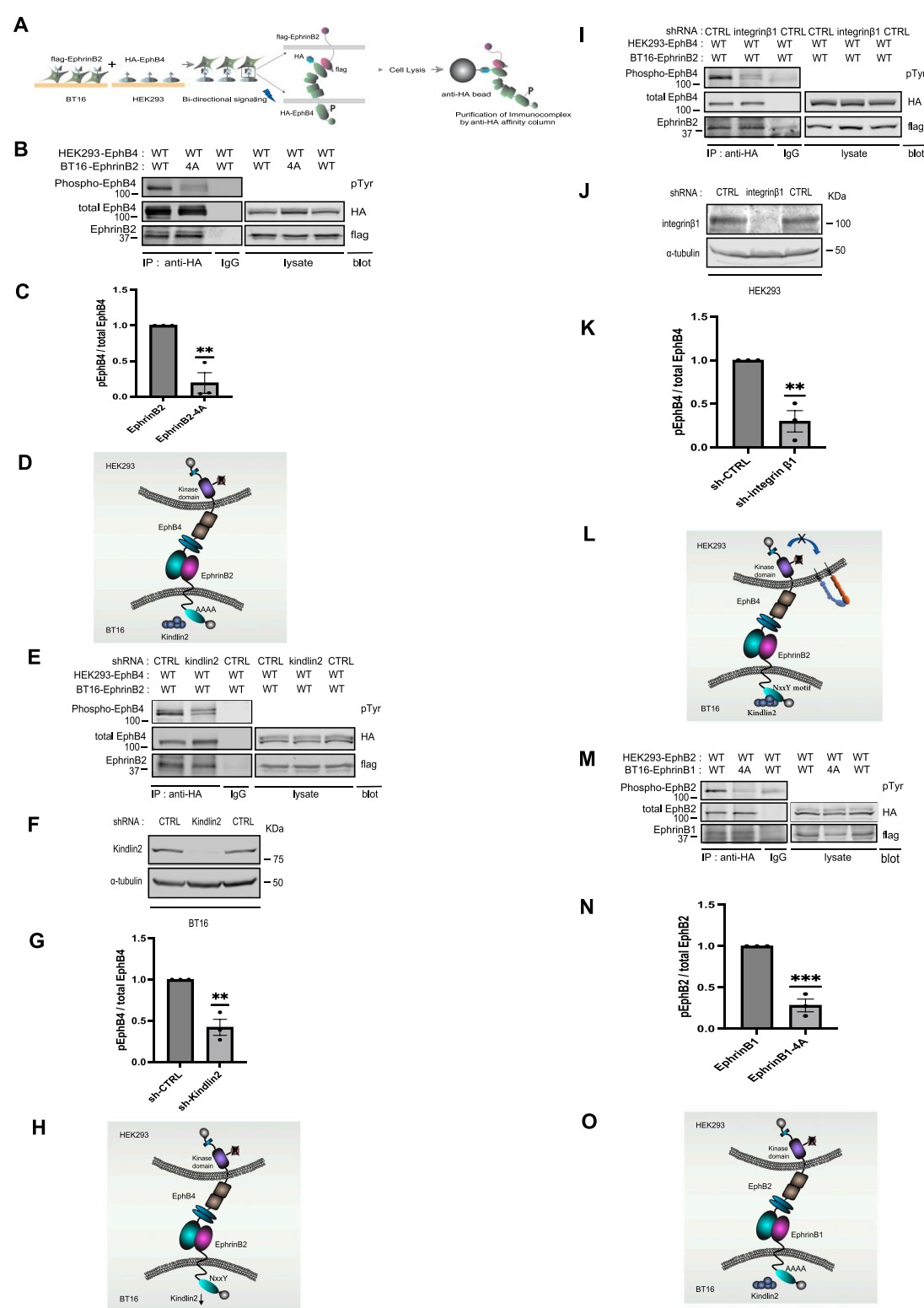

**Figure 6. Kindlin2 binding to the ephrinB2 cytoplasmic tail transduces EphB4-ephrinB2 forward signaling.**
**(A)** Cartoon schematics of two cell populations to assess EphB4 forward signaling. We used BT16 cells that do not express endogenous ephrinB2 for the assay, and kindlin2 is a major kindlin isoform (Fig S4 and https://depmap.org). Stable BT16 cells expressing exogenous flag-ephrinB2 and HEK293 cells expressing HA-EphB4 were generated by lentivirus-mediated gene integration, respectively. The two cell populations were harvested and combined for 2 h. Cell lysates were produced and immunoprecipitated with anti-HA antibody. Bound immunocomplex was separated on SDS–PAGE, and tyrosine phosphorylation of EphB4 was assessed by Western blotting with anti-phosphotyrosine–specific antibody. Cell lysate immunoprecipitated by normal IgG was used as a negative control. **(B)** Kindlin2 binding to the ephrinB2

required to accurately assess the effects of tyrosine phosphorylation on kindlin2 binding.

Earlier genetic studies demonstrated that deletion of the ephrinB2 cytoplasmic tail caused defects in vasculogenesis and angiogenesis during mouse embryogenesis (Wang et al, 1998; Adams et al, 1999; Gerety et al, 1999). Additional studies showed that reverse signaling through the PDZ domain in the ephrinB2 cytoplasmic tail mediates the angiogenesis by regulating VEGF receptor internalization (Sawamiphak et al, 2010; Wang et al, 2010). However, the cytoplasmic domain in the ephrinB2 cytoplasmic tail required for vascular development and precise molecular mechanism remain elusive. Our data reveal that the ephrinB2 NIYY motif in the cytoplasmic tail and its interaction with kindlin2 can regulate arteriovenous segregation and sprouting angiogenesis in zebrafish. Previous works implicated kindlin proteins as an important player in vascular development. Kindlin2 is a crucial regulator for angiogenesis and vascular permeability in the vascular development of mouse and zebrafish (Pluskota et al, 2011). And kindlin3 increases the expression of VEGF in breast cancer cells that plays major roles with Notch in vasculogenesis and angiogenesis (Sossey-Alaoui et al, 2014). EphrinB2 and EphB4 are specifically expressed in arterial and venous endothelial cells, respectively, and bi-directional signaling of ephrinB2 and EphB4 is required for remodeling of primitive capillary networks to distinct arteries and veins (Wang et al, 1998; Herbert et al, 2009; Swift & Weinstein, 2009). Our findings suggest kindlin2 plays a role in linking the EphrinB2/EphB4 pathway to integrin signaling. It is possible that kindlin2 might control integrin-dependent cell migration such as migration of vascular endothelial progenitor cells (angioblasts) to midline in zebrafish. In fact, mutation of *gridlock* that primarily controls the angioblast migration led to the blockade of blood flow and arteriovenous shunt, which resembles the NIYY motif–dependent phenotype in zebrafish embryo (Zhong et al, 2000; Hogan & Schulte-Merker, 2017).

We performed two-cell population experiments to investigate a molecular mechanism of how kindlin2 regulates EphB4/ephrinB2 signaling. It has been suggested that the cytoplasmic tail of ephrinB1 in ephrinB1-expressing cells can affect the signaling processing in EphB2-expressing cells in two-cell populations and that

components that bind the cytoplasmic tail of EphrinB1 other than PDZ-binding domain are required to orchestrate the signaling effects of ephrinB1 on EphB2 in genetic and proteomic studies (Jorgensen et al, 2009). However, molecular players that are responsible for the transcellular signal processing have been unclear. We found that kindlin2 in ephrinB2-expressing cells mediates forward signaling in EphB4-expressing cells, and this process requires integrins in EphB4-expressing cells. We show kindlin2 also transduces EphB2/EphrinB1 forward signaling, suggesting kindlin2-dependent mechanism is conserved in ephrinB family. Because a clustered or multimeric ephrinB2 ligand is required to induce EphB4 activation (Davis et al, 1994), and kindlin is capable of clustering ligand-occupied integrins (Ye et al, 2013), we examined whether kindlin2 promotes ephrinB2 clustering. Indeed, ephrinB2 clustering was inhibited in cells expressing kindlin2-binding mutant as revealed by TIRF microscopy. EphrinB2-bound kindlin2 might promote ephrinB2 clustering by oligomeric nature of kindlin or binding to phosphatidylinositol 4,5-bisphosphate (PIP2) that oligomerizes proteins such as vinculin. Alternatively, kindlin2 might localize ephrinB2 to environment that favors clustering such as membrane lipid raft microdomains where ephrin signaling is restricted. Thus, these results suggest that kindlin2 enables transcellular EphB4 forward signaling by promoting ephrinB2 cluster formation.

These findings provide new insights on coordinated regulation of EphB/ephrinB bi-directional signaling by kindlin2 for arteriovenous development. Considering the highly conserved cytoplasmic tail of ephrinB in various species, our findings implicate a general role for kindlin in the regulation of EphB/ephrinB-related cellular processes in development, and offer anti-angiogenesis target for cancer treatment.

## Materials and Methods

### Plasmids and cell culture

Lentiviral vector constructs expressing HA-EphB4, flag-ephrinB2, flag-ephrinB2 (4A), EGFP-ephrinB2, or EGFP-ephrinB2 (4A) were

---

cytoplasmic tail in BT16 cells regulates in trans EphB4 activation in HEK293 cells. BT16 stable cells expressing flag-ephrinB2 WT or 4A are incubated with HEK293 stable cells expressing HA-EphB4. Cells were lysed, and EphB4/ephrinB2 complex in cell lysates was immunoprecipitated by anti-HA antibody. And bound EphB4 and EphrinB2 were recognized with anti-HA and anti-flag antibodies in Western blot, respectively. The status of tyrosine phosphorylation of EphB4 was analyzed by anti-phosphotyrosine–specific antibody. **(C)** Ratio of tyrosine-phosphorylated EphB4 to total EphB4 was determined by densitometry in Western blot. The data are shown as the mean ± SEM (n = 3). A one-sample *t* test was used. **P < 0.01. **(D)** Cartoon showing disrupting kindlin2 binding to the ephrinB2 cytoplasmic tail in BT16 cells decreases EphB4 activation in HEK293 cells. **(E)** Silencing kindlin2 expression in BT16 cells reduces EphB4 activation in HEK293 cells. **(F)** Endogenous kindlin2 expression in BT16 cells expressing flag-ephrinB2 was silenced by shRNA (F) and co-cultured with HEK293 cells expressing HA-EphB4. Cell lysates were produced and immunoprecipitated by anti-HA antibody to isolate EphB4/ephrinB2 complex. Captured proteins were fractionated on SDS–PAGE and analyzed by Western blot with anti-phosphotyrosine, anti-HA, and anti-flag antibodies, respectively. **(G)** Tyrosine-phosphorylated EphB4/total EphB4 ratio was assessed by densitometry in Western blot. The data are evaluated as the mean ± SEM (n = 3). A one-sample *t* test was used. **P < 0.01. **(H)** Cartoon illustrates that silencing kindlin2 expression in BT16 cells diminishes EphB4 activation in HEK293 cells. **(I)** Silencing integrin expression in HEK293 cells decreases kindlin2-dependent EphB4 activation. **(J)** BT16 stable cells were co-cultured with HEK293 stable cells that were treated with integrin β1 shRNA to knock down endogenous integrin β1 expression (J). After cells were lysed, EphB4/ephrinB2 complex was immunoprecipitated by anti-HA antibody and the status of EphB4 phosphorylation was assessed by anti-phosphotyrosine antibody. **(K)** Tyrosine phosphorylation status of EphB4 was measured by scanning band intensity in Western blot. Bar graph shows the mean ± SEM (n = 3). A one-sample *t* test was used. **P < 0.01. **(L)** Cartoon displays that repressing endogenous integrin β1 expression in HEK293 cells decreases kindlin2-mediated EphB4 activation. **(M)** Kindlin2 regulates EphB2/ephrinB1 forward signaling. BT16 stable cells expressing flag-ephrinB1 were incubated with HEK293 stable cells expressing HA-EphB2. Cell lysates were produced, and EphB2/ephrinB1 complex was immunoprecipitated by anti-HA antibody. And immunoprecipitated proteins were analyzed for the activation status of EphB2 by Western blotting with anti-phosphotyrosine antibody. **(N)** Quantified band intensities are displayed as phosphoEphB4/total EphB4 ratio. The data are presented as the mean ± SEM (n = 3). Statistical analysis used a one-sample *t* test. ***P < 0.001. **(O)** Cartoon illustrating that disrupting kindlin2 binding to ephrinB1 diminishes EphB2 activation in HEK293 cells.

Source data are available for this figure.

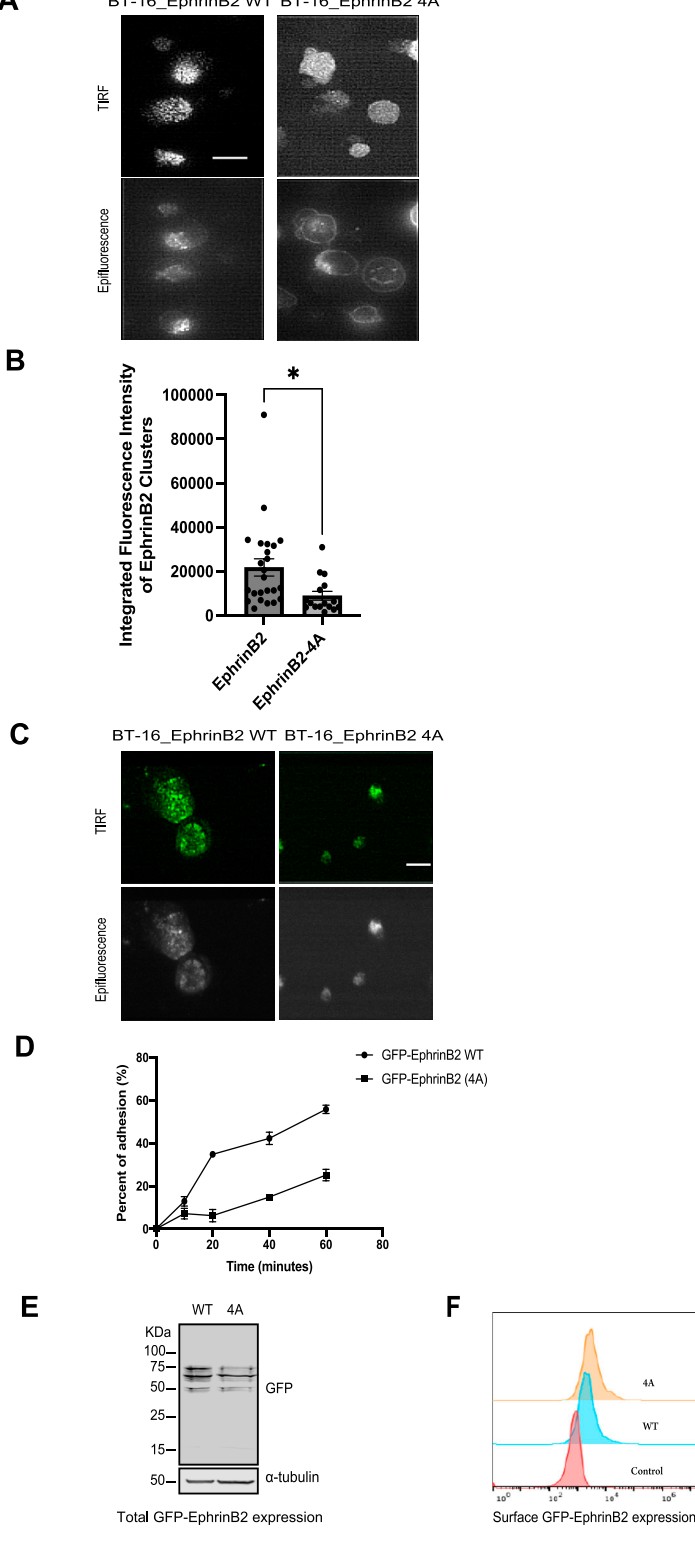

**A** BT-16_EphrinB2 WT   BT-16_EphrinB2 4A

TIRF

Epifluorescence

**B**

Integrated Fluorescence Intensity of EphrinB2 Clusters

*

EphrinB2 / EphrinB2-4A

**C** BT-16_EphrinB2 WT   BT-16_EphrinB2 4A

TIRF

Epifluorescence

**D**

Percent of adhesion (%)

Time (minutes)

● GFP-EphrinB2 WT
■ GFP-EphrinB2 (4A)

**E**

WT  4A
KDa
100—
75—
50—   GFP
25—
15—
50—   α-tubulin

Total GFP-EphrinB2 expression

**F**

4A
WT
Control

Surface GFP-EphrinB2 expression

**Figure 7. Kindlin2 promotes ephrinB2 clustering and cell adhesion.**
**(A)** TIRF and epifluorescence images showing the cluster formation of GFP-ephrinB2 WT or 4A. BT16 cells were transduced with lentivirus particles expressing GFP-ephrinB2 WT or 4A and plated on EphB4-Fc + laminin-coated well for 2 h. Scale bar, 5 μm. **(B)** Fluorescence intensities of ephrinB2 clusters were measured and averaged as described in Experimental Procedures. Error bars indicate the SEM of ephrinB2, n = 24; ephrinB2-4A, n = 16. The data are presented as the mean ± SEM. An unpaired two-tailed $t$ test was used. *$P <$ 0.05. **(C)** TIRF and epifluorescence images showing the cluster formation of GFP-ephrinB2 WT or 4A. BT16 cells were transduced with lentivirus particles expressing GFP-ephrinB2 WT or 4A and plated only on EphB4-Fc–coated well for 2 h. Scale bar, 5 μm. **(D)** Cell culture plates were coated with EphB4-Fc (5 μg/ml), and the free protein-binding sites were blocked with BSA. After cells were plated and incubated for the indicated times, the plates were washed, and quantitative cell adhesion assays were performed as described in Experimental Procedures. **(E)** Total protein expression of GFP-ephrinB2 WT or 4A in BT16 cells was verified by SDS–PAGE of a cell lysate followed by Western blot with anti-GFP antibody. **(F)** Cell surface protein expression on BT16 cells expressing GFP-ephrinB2 WT, 4A, or uninfected control was confirmed by flow cytometric analysis. Cells were harvested and stained with anti-GFP antibody and Alexa Fluor 647–conjugated secondary antibody.
Source data are available for this figure.

made in pLVX-Het1 (Clontech Laboratories, Inc.). HA and flag epitopes were tagged at the NH2-terminus of EphB4 and ephrinB2, respectively, and GFP was fused to the NH2-terminus of ephrinB2. A mammalian expression plasmid encoding full-length DVL2 was

constructed in pEGFP-C1. A cDNA sequence encoding full-length zebrafish kindlin2 was amplified by PCR and cloned into p3×flag-CMV-7.1 (Sigma-Aldrich). Validated shRNAs for ephrinB2 (TRCN0000276587), ephrinB4 (TRCN0000001773), and integrin β1 (TRCN0000275135) were

purchased from Sigma-Aldrich. Lentiviruses were produced in 293T cells co-transfected with the above lentiviral vectors and three packaging plasmids pMDLg/pRRE, pRSV-Rev, and pMD2.G. Bacterial expression constructs encoding the cytoplasmic tail of ephrinB1, ephrinB2, and ephrinB3 were generated in his/Avi-tagged vector in pET15 (Pfaff et al, 1998) by PCR-based cloning. Four alanine mutant (4A) in the cytoplasmic tail was constructed by Q5 site-directed mutagenesis kit (New England Biolabs Inc.) with the following primers: ephrinB1-4A forward (5′-GGCGGCGGCGGCAAGCTTAAGCTACGCAAGCGG-3′) and reverse (5′-CCCCAGAGCCCGGCGGCCGCTGCCGCCAAGGTCTGA GGATCCGGCTGCTAACAA-3′); ephrinB2-4A forward (5′-GGCGGCGGCGG CAAGCTTAAGTACCGGAGGAGA-3′) and reverse (5′-TTGTTAGCAGCCGGA TCCTCAGACCTTGGCGGCAGCGGCCGCCGGGCTCTGCGG-3′); and ephrinB3-4A forward (5′-GGCGGCGGCGGCAAGCTTGCCATGTGTTGGCGG-3′) and reverse (5′-TTGTTAGCAGCCGGATCCTCATACCTTGGCGGCAGCGGCTGGAGG GCTCTGGGG-3′). BT16 cells were a gift from Dr. Houghton at UT Health San Antonio. HEK293 and BT16 cells were grown in DMEM supplemented with 10% (vol/vol) FBS, non-essential amino acids, 1× penicillin/streptomycin, and 2 mM L-glutamine at 37°C in a 5% $CO_2$ incubator. HEK293 stable cell line expressing HA-EphB4 or BT16 stable cell line expressing flag-ephrinB2 or flag-ephrinB2 (4A) was generated by lentiviral transduction of HA-EphB4, or flag-ephrinB2 or flag-ephrinB2 (4A), respectively. Each BT16 stable cell line expressing EGFP-ephrinB2 WT or 4A was also produced by lentiviral infection of EGFP-ephrinB2 WT or 4A.

### Antibodies and reagents

Antibodies used in this study include rabbit anti-ephrinB2 (PA5-35062; Thermo Fisher Scientific), mouse anti-EphB4 (H-10) (sc-365510; Santa Cruz Biotechnology), mouse anti-flag (F3165; Sigma-Aldrich), goat anti-DsRed (sc-33354; Santa Cruz Biotechnology), mouse monoclonal anti-p-Tyr (sc-508; Santa Cruz Biotechnology), mouse anti-paxillin (610051; BD Transduction Laboratories), and anti-α-tubulin (T6074; Sigma-Aldrich). Mouse anti-HA (12CA5) and rabbit anti-EGFP antibodies were developed in the laboratory. Rabbit anti-HA (923501) and anti-DYKDDDDK Tag (L5) Affinity Gel (651502) antibodies were purchased from BioLegend. Secondary Alexa Fluor–labeled antibodies were from Jackson ImmunoResearch. Protein G (L00209) resin was from GenScript. Recombinant mouse EphB4 (extracellular domain residue 16-539)-Fc protein (446-B4-200) was purchased from R&D Systems, Inc.

### BioID and mass spectrometry

RIAM (gene name APBB1IP) was amplified by PCR from a cDNA library from the Mammalian Gene Collection (source sequence BC054516) and cloned in-frame in an entry vector for Gateway cloning (stop codon added). Kindlin2 (gene name FERMT2) was obtained as a closed open reading frame in pENTR221 (sequence reference BC017327). Both ORFs were transferred by Gateway cloning into the pDEST pcDNA5 FLAG BirA* vector for N-terminal tagging, and stable pools of Flp-In T-REx 293 cells (Invitrogen) were generated. Expression was induced alongside biotinylation by the simultaneous addition of tetracycline and biotin for 24 h, and the cells were harvested and processed for mass spectrometry, in biological duplicates, as described previously (Samavarchi-Tehrani

et al, 2018). Samples were analyzed in data-dependent acquisition mode on an AB SCIEX 5600 TripleTOF instrument, and results were jointly analyzed through the search engines Mascot and COMET, followed by iProphet (Shteynberg et al, 2011) analysis. To analyze the specific enrichment of proximal partners in the BioID of RIAM and kindlin2, 12 negative controls and purifications acquired using essentially the same protocol and on the same type of mass spectrometer were first compressed to six virtual controls (selecting, for each prey detected, the six highest spectral counts across the 12 controls to maximize stringency). Significance Analysis of INTeractome (SAINTexpress; [Teo et al, 2014]) was used to define statistically enriched preys, and proteins passing the 1% Bayesian false discovery rate cutoff with at least one of the baits were deemed high-confidence, and visualized using prohits-viz.lunenfeld.ca (Knight et al, 2017).

### Protein purification and pulldown

His/Avi-tagged bacterial expression constructs encoding the cytoplasmic tails of ephrinB1, ephrinB2, ephrinB3, and integrin αIIb or β3 were expressed in *Escherichia coli* (DE3), and the biotinylated recombinant proteins were purified as described previously (Pfaff et al, 1998). A mammalian expression construct encoding his-tagged human full-length kindlin2 was generated in gWIZ vector (Genlantis) by PCR. Recombinant kindlin2 protein was expressed in Expi293F cells and purified according to the instructions of manufacturer using Expi293 Expression System Kit (A14635; Thermo Fisher Scientific). 5 μg of purified cytoplasmic tails immobilized on NeutrAvidin (Pierce Biotechnology) was incubated with EGFP-kindlin2-transfected cell lysate in lysis buffer (50 mM Tris-Cl, 150 mM NaCl, 0.5% NP-40, 0.5% Triton X-100, 0.5 mM $CaCl_2$, 0.5 mM $MgCl_2$, 1 μM calpeptin, protease inhibitor cocktail, and PhosSTOP phosphatase inhibitors, pH 7.4) at 4°C for 1 h. Bound proteins were washed with lysis buffer, fractionated on SDS–PAGE, and recognized by Western blotting with anti-GFP antibody.

### Co-culture and immunoprecipitation

Monolayer cell cultures of HEK293 stable cells expressing HA-EphB4 and BT16 stable cells expressing flag-ephrinB2 or flag-ephrinB2 (4A) were detached from cell culture dish with trypsin–EDTA, respectively. The two-cell populations were mixed and cultured in complete growth medium to allow EphB4/ephrinB2 signaling. Then, the cells were washed and lysed in NP-40 lysis buffer (50 mM Tris–HCl, 150 mM NaCl, 0.5% NP-40, 0.5 mM $CaCl_2$, 0.5 mM $MgCl_2$, 1 μM calpeptin, protease inhibitor cocktail, and PhosSTOP phosphatase inhibitors, pH 7.4) on ice for 15 min and centrifuged at 18,213*g* at 4°C for 15 min. Protein concentration of the supernatant was determined using a BCA assay (Thermo Fisher Scientific). An equal amount of the soluble lysate was immunoprecipitated using designated primary antibodies with protein G resin (GenScript). Bound immunocomplexes were washed, separated by SDS–PAGE, and probed with indicated primary and secondary antibodies.

### FACS analysis of the activation status of integrin

The activation status of integrin in HEK293 stable cells expressing constitutively active integrin αIIb(R995A)β3 was measured by the

binding of the ligand-mimetic anti-$\alpha IIb\beta 3$ monoclonal antibody PAC-1 in two-color flow cytometric assays as described previously (Tadokoro et al, 2003). The expression of endogenous ephrinB2 or EphB4 in HEK293 ($\alpha IIb[R995A]\beta 3$) cells was silenced by lentiviral infection of shRNA, and the cells were transfected with shRNA-resistant ephrinB2 or EphB4 constructs, respectively. After 24 h, cells were suspended and stained with PAC-1. Bound PAC-1 was assessed on a BD Accuri C6 flow cytometer (BD Biosciences). Activation was quantified as an activation index calculated as ($F$-$Fo$)/($Fmax$-$Fo$), where $F$ is the mean fluorescence intensity (MFI) of PAC-1 binding, $Fo$ is the MFI of PAC-1 binding in the presence of the $\alpha IIb\beta 3$ antagonist Integrilin (10 $\mu$M), and $Fmax$ is the MFI of PAC-1 binding in the presence of $\alpha IIb\beta 3$-activating mAb anti-LIBS6 (2 $\mu$M). The activation state of endogenous integrin $\beta 1$ in HEK293 cells was assessed by measuring the binding of a 9EG7 antibody that activates integrin $\beta 1$. The expression of endogenous ephrinB2 or EphB4 in HEK293 cells was silenced by lentivirus-based RNAi. After selection with puromycin (1 $\mu$g/ml), cells were suspended and stained with 9EG7 (5 $\mu$g/ml). Bound 9EG7 was measured by FACS analysis.

### TIRF microscopy

TIRF imaging was performed on an Oxford Nanoimager (ONI) microscope. The ONI is equipped with a scientific complementary metal–oxide–semiconductor camera, four laser lines (405, 473, 532, and 640 nm), and a UAPON100XOTIRF1.49NA oil objective (Olympus). Two fluorescence channels were separated with dichroic (560LP) and emission (525/50 and 575-616.5) filters. High-resolution TIRF pictures were also acquired by Applied Precision OMX Super-Resolution (GE Healthcare) equipped with 100× (1.40 NA) oil objective (Olympus), and images were deconvoluted with SoftWoRx software (Applied Precision).

For co-localization analysis (Fig 3A and B), NIH3T3 cells expressing mCherry-kindlin2 with EGFP-ephrinB2 or with EGFP-ephrinB2 (4A) were plated on $\mu$-Slide eight-well (ibidi) coated with laminin (5 $\mu$g/ml) (L2020; Sigma-Aldrich). 24 h after cell seeding, the cells were fixed with 3.7% formaldehyde for 15 min at room temperature and washed with PBS. Subsequently, the fixed cells were used for TIRF imaging. For clustering analysis, BT16 cells expressing EGFP-ephrinB2 or EGFP-ephrinB2 (4A) were plated on $\mu$-Slide eight-well (ibidi) coated with EphB4-Fc (5 $\mu$g/ml) and laminin (5 $\mu$g/ml). 3 h after cell seeding, the cells were fixed with 3.7% formaldehyde for 15 min at room temperature and washed with PBS. Subsequently, the fixed cells were used for TIRF imaging.

### Image analysis and quantification

Raw TIRF images of stable EGFP-positive ephrinB2 clusters in BT16 cells were thresholded by ImageJ. The area- and background-corrected pixel intensities of ephrinB2 clusters were measured using Analyze Particles tool of ImageJ. Integrated fluorescence intensity was depicted as mean fluorescence intensity × area. A total of 24 fields for ephrinB2 WT and 16 fields for ephrinB2 (4A) were quantified.

### Cell adhesion assay

24-well tissue culture plate was coated with recombinant mouse EphB4-Fc (5 $\mu$g/ml) for 1 h at 37°C. The wells were blocked with 1%

heat-inactivated BSA in PBS for 30 min at room temperature and washed three times with PBS. Growing BT16 cells expressing EGFP-ephrinB2 WT or 4A were harvested and washed with PBS. The cells ($0.5 \times 10^5$) were seeded into wells and incubated at 37°C in a 5% $CO_2$ incubator for the indicated times. Non-adherent cells were washed away with three vigorous PBS washes, and the adhered cells were suspended and quantified by flow cytometer.

### Zebrafish lines and husbandry

$Tg(fli1:EGFP)^{y1}$ zebrafish were maintained with the approval of the Institutional Animal Care and Use Committee of the University of California, San Diego.

### *Plasmids and mRNA*

Zebrafish ephrinb2a (gene name efnb2a) was amplified from 2 dpf cDNA with the following primers: efnb2a forward (5'-ATCTCTA-GAATGGGCGACTCTTTGTGGAGAT-3') and reverse (5'-ATCAAGCTTGTT-CATCGAGGGGCATGTGA-3'). The fragment was then inserted into pcDNA3.1(−) vector between XbaI and HindIII. A 4A mutant construct was generated using site-directed mutagenesis (New England Biolabs, Inc.) with the following primers: forward (5'-CAGAGCC-CAGCAGCCGCCGCTGCCAAGGTGTGAAAA-3') and reverse (5'-TGGCG GCATTTCCTGTACGAT-3'). Both ephrinb2a (WT) and ephrinb2a (4A) plasmids were digested by HindIII and then purified by column (Macherey-Nagel Inc.) as a template for in vitro transcription. Capped mRNA was produced with mMESSAGE mMACHINE T7 ULTRA Transcription Kit (AM1345; Thermo Fisher Scientific) and purified through lithium chloride precipitation described in the same kit. The concentration of mRNA was measured by NanoDrop 1000 Spectrophotometer (Thermo Fisher Scientific), and their quality was confirmed by electrophoresis in 1% (wt/vol) agarose gel. 500 pg of mRNA was injected into one-cell-stage embryos.

### *Airyscan imaging*

Injected embryos for imaging were anesthetized with 0.016% tricaine (3-amino benzoic acid ethyl ester; Sigma-Aldrich) and embedded using 1% low melting point agarose (16520050; Invitrogen). Imaging was performed with a Zeiss 880 Airyscan confocal microscope under the standard mode.

## Supplementary Information

## Acknowledgements

We thank Mark H Ginsberg for helpful discussion and Zhen-Yuan Lin for help with BioID and mass spectrometry. A-C Gingras is supported by the Canadian Institutes of Health Research (CIHR FDN 143301) and a Canada Research Chair, Tier 1, in Functional Proteomics. The equipment used for proteomics is housed in the Network Biology Collaborative Centre at the Lunenfeld-Tanenbaum Research Institute, a facility supported by Canada Foundation for Innovation funding, by the Ontarian Government, and by Genome Canada and Ontario Genomics (OGI-139). This work was supported by the

National Institutes of Health grants R35 HL145241 (to K Ley) and R35 HL139947 (to Mark H Ginsberg). We thank Jennifer Santini for microscopy technical assistance. We also acknowledge resources provided by the UCSD Microscopy Core (NINDS P30 NS047101).

## Author Contributions

W Li: data curation, formal analysis, validation, investigation, visualization, methodology, and writing—original draft.
L Wen: data curation, formal analysis, validation, investigation, visualization, methodology, and writing—original draft.
B Rathod: data curation, validation, and methodology.
A-C Gingras: resources, data curation, formal analysis, supervision, validation, and methodology.
K Ley: resources, data curation, supervision, validation, and methodology.
H-S Lee: conceptualization, resources, data curation, formal analysis, supervision, funding acquisition, validation, investigation, visualization, methodology, project administration, and writing—original draft.

## Conflict of Interest Statement

The authors declare that they have no conflict of interest.

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
