## [Reviewer comments · Life Science Alliance]

Life Science Alliance

Kindlin2 Enables EphB/EphrinB Bi-directional Signaling to Support Vascular Development

Wenqing Li, Lai Wen, Bhavisha Rathod, Anne-Claude Gingras, Klaus Ley, and Ho-Sup Lee

DOI: <https://doi.org/10.26508/lsa.202201800>

Corresponding author(s): Ho-Sup Lee, University of California, San Diego

Review Timeline:

Submission Date:	2022-11-03
Editorial Decision:	2022-11-04
Revision Received:	2022-12-01
Editorial Decision:	2022-12-02
Revision Received:	2022-12-09
Accepted:	2022-12-12

Transaction Report:

Please note that the manuscript was previously reviewed at another journal and the reports were taken into account in the decision-making process at Life Science Alliance.

Reviewer #1 Review

Comments to the Authors (Required):

The authors have performed a number of new experiments in response to the reviewers' comments, substantially improving the manuscript. However, there are still significant concerns.

The major concern is that the authors conclude that the NIYY motif of ephrin-B2 (which is reminiscent of the NXXY motif in integrins) mediates the interaction with kindlin2. This motif is mentioned throughout the manuscript. That the kindlin2 binding motif might be the same in integrins and ephrinB2 was the hypothesis to be tested, and the data obtained do not support it. The new Fig. 2E,F shows that the N328A mutation does not affect kindlin2 binding. This suggests that "N" is not part of the motif. I329 was not mutated, so its contribution to kindlin2 binding remains unknown. It appears that the kindlin2 binding motif might be IYY (not NIYY), provided that experiments that should be performed confirm that the I329A mutation also impairs kindlin2 binding. In fact, the domain of kindlin2 mediating ephrinB2 binding is different from the domain mediating integrin binding (as stated on page 27), so there is no reason to think that the same binding motif should be involved. In fact, even the upstream serine (S325 in ephrinB2), which is important for integrin binding, is not important for ephrinB2 interaction with kindlin2.

The authors have not addressed whether phosphorylation of Y330 and/or Y331 in ephrinB2 regulates kindlin-2 binding. On page 27, the authors suggest that "tyrosine phosphorylation of Y331 (NIYpY) might minimally affect kindlin2 binding". However, the Y331A mutation, which they use as the basis for their suggestion, is not useful to draw conclusions on the effects of phosphorylation. EphrinB2 synthetic peptides in which Y330, Y331 or both are replaced by phosphotyrosine should be used in kindlin2 binding assays.

Fig. S3 and page 18. No control cells without ephrinB2 expression are shown. Therefore, the evidence presented does not support the statement on page 18 that "Expression of EGFP-ephrinB2 did not alter the formation or distribution of focal adhesions...".

Minor points

The labels in Fig. 5 C-H appears to be incomplete. Should they be efnb2a for C,F and efnb2a(4a) for D,G?

Fig. S2 is still confusing. The legend explains some of the abbreviations below the dot plots but not others. For example, what does HPA mean? What does HP mean? What do the numbers near the circles with a black outline mean? Which circles do the GO terms in the lower part of the figure correspond to?

Reviewer #2 Review

Comments to the Authors (Required):

The authors have revised and improved their manuscript in response to initial reviews but concerns remain.

The addition of studies with purified proteins, in combination with the pull-down assays from cell lysate, co-IP experiments and the initial BiolD results increase confidence in the direct kindlin-ephrin interaction - although detailed quantitative characterization of the affinity of the interaction is still lacking.

It remains somewhat perplexing that strong disruption of binding requires mutation of all 4 residues in the NIYY motif but understanding this probably will require additional information on the binding sites in kindlin that seems to be beyond the scope of this manuscript. However, in the discussion the authors mention that ephrin probably does not bind at the integrin-binding site on kindlin - this is an important finding and it would be helpful to present these results early in the results section of the manuscript. The initial assumption and selection of the NIYY motif seemed to be based on the similarity to the integrin sequence hence at an early stage of the manuscript readers are left with the impression that ephrin binding to kindlin will occur at the integrin-binding site. Presenting the data that refute this idea early will help understanding of the story.

The co-localization data have been strengthened but the new supplementary figure (Fig S3) suggests that the wild-type but not 4A-mutant ephrin is in focal adhesions. Is correlation of ephrin with paxillin also strongly impacted by the 4A mutation? The loss of mutant ephrin from focal adhesions should be commented on.

My major unaddressed concern is that throughout the manuscript testing the significance of the kindlin-ephrin interaction relies entirely on the 4A mutant or on knockdown studies. The specificity of each of these approaches for selectively disrupting kindlin-ephrin interactions is not established - knockdown of kindlin will impact all kindlin interactions and the 4A mutation may block both ephrin phosphorylation and binding to other known or unknown partners. Consequently, many of the conclusions in the text need to be tempered. The authors' results establish that the NIYY motif is important for ephrin functions and for impacts on integrins but they cannot always say that this is due to disturbed kindlin binding. At a minimum, this needs to be more clearly acknowledged and explicitly described.

Reviewer #1 Review

Comments to the Authors (Required):

This manuscript reports a novel signaling interaction between ephrin-B2 and kindlin2, which induces ephrin-B2 clustering thus promoting the ability of ephrin-B2 to both mediate reverse signals and stimulate forward signaling through EphB receptors. The proposed mechanism likely also applies to the other two ephrin-B ligands, ephrin-B1 and ephrin-B3 and to the other two members of the kindlin family. The authors further propose a physiological role of the ephrin-B2-kindlin2 interaction in promoting EphB4 kinase-dependent integrin activation and enabling blood vessel morphogenesis using zebrafish embryos as a model. The findings are interesting and clearly presented, and represent a significant advance in our understanding mechanisms of EphB/ephrinB signaling.

The following issues should be addressed to further improve the manuscript.

The existing literature on the regulation of integrin-mediated adhesion by ephrins/Eph receptors should be discussed in more depth (including at least mentioning some references to articles reporting opposite effects on integrin activity). Zang et al 2015 Nat Comm 6:6625 seems particularly relevant to the findings reported in the manuscript. Other potentially relevant literature includes Foo et al 2005 Cell 124:161; Meyer et al 2005 Int J Oncol 27:1197; Hamada et al 2003 Arterioscler Thromb Vasc Biol 23:190; Zou et al 1999 PNAS 96:13813; Miao et al 2005 JBC 280:923.

N (the number of experiments, samples) should be indicated in the figure legends. In the graphs, where appropriate, individual datapoints should be shown in addition to the averages and standard errors.

In Fig. 6C,G,K,N, one sample t-test should be used to determine statistical significance for the difference from 1 (the normalized value of the controls, which do not have an error).

In the Methods, please indicate whether the HA, flag, and EGFP tags are at the N-terminus or C-terminus of ephrinB2 or EphB4. Please also indicate how the cells were "harvested" for experiments.

Page 17, second paragraph. There are previous publications on the effects of ephrinB2 on cell substrate adhesion, which should be referenced here (see comment above on literature references).

Pages 19-20. The sentence "further examination of the vessels showed that ephrinb2a overexpressing led to a narrow aorta, shorter intersegmental vessel (Figure 5C) or arterial-venous shunt (Figure 5E)." is repeated twice.

Page 21. The finding that integrin beta1 is required for EphB4 activation in the EphB4 expressing cells should be better explained in the context of the other findings in the manuscript. Might kindlin2 also play an important role in the EphB4-expressing cells?

Lines 2-3 on page 22. "In contrast, soluble monomeric ligands do not induce the Eph receptor autophosphorylation (Davis et al., 5 1994)." This is somewhat controversial, at least for EphA receptors, since soluble monomeric versions of ephrinAs have been reported to activate EphA2 (and other EphA receptors). See, for example, Wykosky et al 2008 Oncogene 58:7260.

Page 22. The implications of the difference between adhesion of ephrinB2-expressing cells to a substrate coated with EphB4-Fc + laminin (Fig. 7A,C) or only with EphB4-Fc (Fig. 7C, D) should be better explained.

Page 25, second paragraph. The second tyrosine in the ephrinB2 NIYY motif is also a phosphorylation site (phosphosite.org). Could phosphorylation of this second tyrosine affect interaction with kindlin2? If so, the sentence "kindlin2 binding to ephrinB2 may not be regulated by tyrosine phosphorylation" might not be accurate.

In the Fig. 1 legend, or elsewhere in the manuscript, it should be stated that FERMT2 is the gene name for kindlin2 and APBB1IP is the gene name for RIAM.

Fig. 2B,C. The asterisks are difficult to see in this and other figures where the bands are very dark.

In Fig. 3A, GFP should be indicated at the top of the first two panels (next to mCherry-kindlin2 and Merge).

In Fig. 5, it would be desirable to also show a WT embryo for comparison.

The Fig. 6A legend seems to imply that in this experiment interactions occur between cells in suspension ("the two cell populations were harvested and combined for 2 hours"), but the figure shows that the HEK293 cells are attached to the plate. This point should be clarified.

The Fig. S2 legend does not provide information sufficient to fully understand the figure.

Movie 1 should show a normal zebrafish embryo, so that the abnormal features in the subsequent movies can be better appreciated. Important features of the supplementary movies should be labeled.

Reviewer #2 Review

Comments to the Authors (Required):

This manuscript reports a kindlin-ephrinB interaction and suggests that this interaction is important for bidirectional signaling through Eph-ephrin receptors and through integrins and that it contributes to vascular development. The topic is interesting and, as far as I am aware, novel. Potential roles for kindlin in ephrin signaling opens new areas of study and understanding and may have implication for results with kindlin mutants that are currently interpreted in the context of alterations in integrin function. As such, the topic is likely to be of interest for readers of this journal. My major concerns are the lack of detailed characterization of the kindlin-ephrinB interaction and the heavy reliance on the quadruple 4A ephrinB mutant rather than more selective mutants for all functional conclusions. Currently, the data show that the 4A mutant impairs ephrinB function but whether this is due to defective ephrin-kindlin interactions or to the loss of some other interaction, perhaps loss of phosphorylation, is much less clear. As such, the conclusions that kindlin binding is important are weakly supported.

Specific comments:

BioID links kindlin-2 to ephrinB and pulldown assays suggest interactions between kindlin-2 and a NIYY motif in the ephrinB tail. This interaction is central to the conclusions of this manuscript and warrants more detailed characterization. Although suggestive, the proximity ligation data and pull-down from lysates do not establish a direct interaction between ephrinB and kindlin-2. Experiments with purified proteins would help here and would allow comparison of affinities between integrin-binding and ephrinB binding.

In integrin-kindlin interactions individual point mutations are sufficient to disrupt binding (mutating all 4 residues in the NxxY motif is not required). Which of the 4 ephrin residues are important here? Likewise, the S/T residues preceding the NxxY motif are important for integrin-kindlin interactions is this the case for ephrin too?

Kindlin mutants defective in binding to integrin NxxY motifs are known - are these mutants also defective in binding ephrins? I.e., does kindlin engage ephrin in a manner analogous to how it engages integrin? Data described in the discussion section seems to address this question but it would be more appropriate to include a proper description of these data in the results section.

The data suggesting that ephrins engage kindlins in a manner distinct from integrins raises additional questions. The authors selected the NIYY motif for mutagenesis based on its similarity to the kindlin-binding site in integrins but if a different kindlin site is involved there is little basis for comparing the integrin and ephrin interactions with kindlin. Conclusions about the likely impact of phosphorylation of the NIYY motif on kindlin binding can then not be made based on integrin results. Indeed, it seem possible that the functional effects of the 4A mutation may be related to the loss of this phosphorylation site rather than a loss of kindlin binding.

Expression controls for wild-type and mutant proteins should include assessment of cell-surface proteins (by flow cytometry) not just immunoblotting experiments as transmembrane proteins such as ephrins need to correctly traffic to the cell surface to function and loss of normal trafficking or increased internalization could account for changes in the functions of the 4A mutant.

The colocalization data shown in Fig 3A suggest that ephrin expression may alter the localization of mCherry kindlin. The patterns of kindlin localization in the GFP-ephrinB2 cells looks different from that in the 4A cells. In the 4A cells punctate kindlin localization resembles a peripheral and central focal adhesion pattern but in the ephrinB2 cells only peripheral staining is seen - are these all the focal adhesions or is kindlin excluded from some adhesions in the presence of ephrinB2 perhaps due to competition with integrin for binding. Co-staining with an additional focal adhesion marker would be helpful. Does ephrin B expression alter the formation or distribution of focal adhesions?

The profound inhibition of integrin activation by ephrinB2 knockdown in 293 cells is striking and somewhat surprising. Is this true for other integrins, e.g. endogenous alpha5beta1 integrins, or only the engineered mutant integrin used here? Why was this mutant integrin chosen rather than endogenous integrins or wild-type integrins?

The presence of both ephrinB2 and its receptor EphB4 in the cells used complicates molecular analysis of the signaling but the knockdown data suggest that loss of either component inhibits integrin activation. Are integrins inactive in other cells lacking EphrinB? As kindlin binding to integrin is required for integrin activation and the integrin-binding and ephrin-binding sites on kindlin are likely to overlap how might kindlin-ephrin interactions contribute to integrin activation?

Over-expression of ephrinB2a but not the NIYY (4A) mutant disrupts blood circulation. Here again, it is important to control for the surface expression of the mutant proteins.

Reviewer #3 Review

Comments to the Authors (Required):

The authors report that the interaction of kindlin2 with the cytoplasmic tail motif NIYY of ephrinB1 and ephrinB2 is required for bidirectional ephrinB1/2 signaling in different mammalian cells and endothelial cells of developing zebrafish. This is the first report of an interaction between kindlin2 and ephrinB1/2 and characterization in vitro and in vivo. Therefore, it is of utmost importance to provide convincing results. The major concern is that almost all experiments are carried out with overexpressed proteins.

- the ephrinB2 cytoplasmic tail harbors the SPANIYY motif that serves as kindlin binding site. Mutation of NIYY reduces but does not abolish kindlin interaction. This is astonishing and begs for a better characterization of the interaction site. Such a characterization should be done with recombinant kindlin rather than with cell lysates that demonstrate an interaction but not necessarily a direct one.

- the different cell lines used in the study were seeded on laminin (laminin-1-1-1?). The signals obtained after mCherry-kindlin-2 and GFP-ephrinB2 overexpression resembles an integrin adhesion-like staining pattern that becomes disrupted upon expression of GFP-ephrinB2 (4A). Do wild type GFP-ephrinB2 and kindlin-2 colocalize with integrins and is the integrin colocalization of kindlin2 with integrins lost in GFP-ephrinB2 (4A)-expressing cells? Why does the mutation of NIYY to 4A lead to the translocation of ephrinB2 and kindlin from the cell periphery to the cell center? Why are wildtype ephrinB2 and ephrinB2-4A in BT16 cells (Fig. 7) signals so similar in number and distribution? In contrast to Figure 3, EphrinB2 WT is not present in the periphery of BT16 cells? Is this due to the presence of EphB4?

- the zebrafish experiments and results are far from clear, and moreover, lack wildtype controls. The conclusions seem to be an overinterpretation!

Overall, the study raises more questions than it answers. The findings are too preliminary and require careful confirmation.

November 4, 2022

Re: Life Science Alliance manuscript #LSA-2022-01800-T

Dr. Ho-Sup Lee
University of California San Diego Medical Center
Medicine
9500 Gilman Dr.
La Jolla, CA 92093

Dear Dr. Lee,

Thank you for submitting your manuscript entitled "Kindlin2 Enables EphB/EphrinB Bi-directional Signaling to Support Vascular Development" to Life Science Alliance. We invite you to submit a revised manuscript addressing the following Reviewer comments:

- To address the key remaining concerns of Reviewers 1 and 2, tone down the claim that the NIYY motif of ephrin-B2 mediates the interaction with kindlin2. Also address Reviewer 1's minor points.

Thank you for this interesting contribution to Life Science Alliance. We are looking forward to receiving your revised manuscript.

Sincerely,

B. MANUSCRIPT ORGANIZATION AND FORMATTING:

Reviewer #1 (Comments to the Authors (Required)):

The authors have performed a number of new experiments in response to the reviewers' comments, substantially improving the manuscript. However, there are still significant concerns.

The major concern is that the authors conclude that the NIYY motif of ephrin-B2 (which is reminiscent of the NXXY motif in integrins) mediates the interaction with kindlin2. This motif is mentioned throughout the manuscript. That the kindlin2 binding motif might be the same in integrins and ephrinB2 was the hypothesis to be tested, and the data obtained do not support it. The new Fig. 2E,F shows that the N328A mutation does not affect kindlin2 binding. This suggests that "N" is not part of the motif. I329 was not mutated, so its contribution to kindlin2 binding remains unknown. It appears that the kindlin2 binding motif might be IYY (not NIYY), provided that experiments that should be performed confirm that the I329A mutation also impairs kindlin2 binding. In fact, the domain of kindlin2 mediating ephrinB2 binding is different from the domain mediating integrin binding (as stated on page 27), so there is no reason to think that the same binding motif should be involved. In fact, even the upstream serine (S325 in ephrinB2), which is important for integrin binding, is not important for ephrinB2 interaction with kindlin2.

Our data show IYY motif is required for kindlin2-binding. Therefore, we tone-downed NIYY to IYY in NxxY motif for kindlin2-binding.

The authors have not addressed whether phosphorylation of Y330 and/or Y331 in ephrinB2 regulates kindlin-2 binding. On page 27, the authors suggest that "tyrosine phosphorylation of Y331 (NIYpY) might minimally affect kindlin2 binding". However, the Y331A mutation, which they use as the basis for their suggestion, is not useful to draw conclusions on the effects of phosphorylation. EphrinB2 synthetic peptides in which Y330, Y331 or both are replaced by phosphotyrosine should be used in kindlin2 binding assays.

This requires to synthesize a peptide that is biotinylated and tyrosine-phosphorylated at Y333 (NIYpY), which will surpass revision time limit. However, We agree the importance of phosphorylation in NIYY motif for kindlin2-binding and downstream signaling, so we are planning to test it in a next project.

Fig. S3 and page 18. No control cells without ephrinB2 expression are shown. Therefore, the evidence presented does not support the statement on page 18 that "Expression of EGFP-ephrinB2 did not alter the formation or distribution of

focal adhesions...".

We performed a control experiment where ephrinB2 wildtype is not expressed in cells. Cells were transduced with lentivirus particles expressing GFP alone or GFP-ephrinB2 4A, respectively, and stained for focal adhesion marker, paxillin. We examined focal adhesions by TIRF microscopy (Reviewer Fig. 1). Expression of GFP alone or ephrinB2 mutant (4A) does not alter the formation or distribution of focal adhesions.

Reviewer Figure 1. NIH 3T3 cells were transduced by lentivirus particles expressing GFP-ephrinB2 4A or GFP alone, respectively. The cells were plated on laminin coated wells and stained with anti-paxillin antibody. Localization of the proteins was visualized by TIRF microscope. Scale bar, 5 µm.

Minor points

The labels in Fig. 5 C-H appears to be incomplete. Should they be efnb2a for C,F and efnb2a(4a) for D,G?

We fixed errors in the figure 5.

Fig. S2 is still confusing. The legend explains some of the abbreviations below the dot plots but not others. For example, what does HPA mean? What does HP mean? What do the numbers near the circles with a black outline mean? Which circles do the GO terms in the lower part of the figure correspond to?

We described the figure legends in more detail (page 39, 40).

Reviewer #2 (Comments to the Authors (Required)):

The authors have revised and improved their manuscript in response to initial reviews but concerns remain.

The addition of studies with purified proteins, in combination with the pull-down assays from cell lysate, co-IP experiments and the initial BioID results increase confidence in the direct kindlin-ephrin interaction - although detailed quantitative characterization of the affinity of the interaction is still lacking.

It remains somewhat perplexing that strong disruption of binding requires mutation of all 4 residues in the NIYY motif but understanding this probably will require additional information on the binding sites in kindlin that seems to be beyond the scope of this manuscript. However, in the discussion the authors mention that ephrin probably does not bind at the integrin-binding site on kindlin - this is an important finding and it would be helpful to present these results early in the results section of the manuscript. The initial assumption and selection of the NIYY motif seemed to be based on the similarity to the integrin sequence hence at an early stage of the manuscript readers are left with the impression that ephrin binding to kindlin will occur at the integrin-binding site. Presenting the data that refute this idea early will help understanding of the story.

We moved Fig. S7 to regular Fig. 2G, H (page 17, 18).

The co-localization data have been strengthened but the new supplementary figure (Fig S3) suggests that the wild-type but not 4A-mutant ephrin is in focal adhesions. Is correlation of ephrin with paxillin also strongly impacted by the 4A mutation? The loss of mutant ephrin from focal adhesions should be commented on.

Our data show 4A-mutant is not present in focal adhesions (Fig. 3A, S3). This result suggests localization of ephrinB2 in paxillin-containing focal adhesion is likely dependent upon binding to kindlin2. We added comments on the loss of mutant ephrin in the results (page 19).

My major unaddressed concern is that throughout the manuscript testing the

significance of the kindlin-ephrin interaction relies entirely on the 4A mutant or on knockdown studies. The specificity of each of these approaches for selectively disrupting kindlin-ephrin interactions is not established - knockdown of kindlin will impact all kindlin interactions and the 4A mutation may block both ephrin phosphorylation and binding to other known or unknown partners. Consequently, many of the conclusions in the text need to be tempered. The authors' results establish that the NIYY motif is important for ephrin functions and for impacts on integrins but they cannot always say that this is due to disturbed kindlin binding. At a minimum, this needs to be more clearly acknowledged and explicitly described.

We agree that 4A mutation might block other binding partners to interact with ephrinB2 cytoplasmic tail. That is why we tested DVL2 that contains PDZ domain and binds PDZ-binding motif in the cytoplasmic tail (Fig. 2C). The PDZ-binding motif is partially overlapped with kindlin2-binding NxxY motif (Fig. 2A). DVL2 is able to bind 4A mutant and ephrinB2 wildtype equally well. This result shows 4A mutation did not disrupt the ability of DVL2 to bind ephrinB2 tail.

December 2, 2022

RE: Life Science Alliance Manuscript #LSA-2022-01800-TR

Dr. Ho-Sup Lee
University of California, San Diego
Medicine
9500 Gilman Dr.
La Jolla, CA 92093

Dear Dr. Lee,

Thank you for submitting your revised manuscript entitled "Kindlin2 Enables EphB/EphrinB Bi-directional Signaling to Support Vascular Development". We would be happy to publish your paper in Life Science Alliance pending final revisions necessary to meet our formatting guidelines.

- please upload your manuscript text as an editable doc file
- please upload both your main and supplementary figures as single files
- please upload your main figures as single page files; these will be displayed in-line in the HTML version of your paper, so please provide them as single page files (Figures 2, 4, 6, 7 currently span multiple pages); we do not have a limit on the number of main figures and these can be split if necessary for space
- please add the Author Contributions to the main manuscript text
- please consult our manuscript preparation guidelines <https://www.life-science-alliance.org/manuscript-prep> and make sure your manuscript sections are in the correct order
- please use the [10 author names, et al.] format in your references (i.e. limit the author names to the first 10)
- please remove the Conclusion text from the Graphical Abstract, and be sure to label this file as a Graphical Abstract. It is currently labeled as Cover Art.

Figure Check:

- please add sizes next to all blots
- the blots in Figure S6 appear pixelated

A. FINAL FILES:

B. MANUSCRIPT ORGANIZATION AND FORMATTING:

Sincerely,

December 12, 2022

RE: Life Science Alliance Manuscript #LSA-2022-01800-TRR

Dr. Ho-Sup Lee
University of California, San Diego
Medicine
9500 Gilman Dr.
La Jolla, CA 92093

Dear Dr. Lee,

Thank you for submitting your Research Article entitled "Kindlin2 Enables EphB/EphrinB Bi-directional Signaling to Support Vascular Development". It is a pleasure to let you know that your manuscript is now accepted for publication in Life Science Alliance. Congratulations on this interesting work.

DISTRIBUTION OF MATERIALS:

Again, congratulations on a very nice paper. I hope you found the review process to be constructive and are pleased with how the manuscript was handled editorially. We look forward to future exciting submissions from your lab.

Sincerely,
